# PC-PG: Policy Cover Directed Exploration for Provable Policy Gradient Learning

**Alekh Agarwal**
Microsoft Research, Redmond
alekha@microsoft.com

**Sham Kakade**
University of Washington and Microsoft Research NYC
sham@cs.washington.edu

**Mikael Henaff**
Facebook AI Research
mikaelhenaff@fb.com

**Wen Sun**
Cornell University
ws455@cornell.edu

## Abstract

Direct policy gradient methods for reinforcement learning are a successful approach for a variety of reasons: they are model free, they directly optimize the performance metric of interest, and they allow for richly parameterized policies. Their primary drawback is that, by being local in nature, they fail to adequately explore the environment. In contrast, while model-based approaches and Q-learning can, at least in theory, directly handle exploration through the use of optimism, their ability to handle model misspecification and function approximation is far less evident. This work introduces the the POLICY COVER GUIDED POLICY GRADIENT (PC-PG) algorithm, which provably balances the exploration vs. exploitation tradeoff using an ensemble of learned policies (the policy cover). PC-PG enjoys polynomial sample complexity and run time for both tabular MDPs and, more generally, linear MDPs in an infinite dimensional RKHS. Furthermore, PC-PG also has strong guarantees under model misspecification that go beyond the standard worst case $\ell_\infty$ assumptions; these include approximation guarantees for state aggregation under an average case error assumption, along with guarantees under a more general assumption where the approximation error under distribution shift is controlled. We complement the theory with empirical evaluation across a variety of domains in both reward-free and reward-driven settings.

## 1 Introduction

Policy gradient methods are a successful class of Reinforcement Learning (RL) methods, as they are amenable to parametric policy classes, including neural policies [55, 56]), and they directly optimizing the cost function of interest. While these methods have a long history in the RL literature [33, 37, 62, 68], only recently have their theoretical convergence properties been established: roughly when the objective function has wide coverage over the state space, global convergence is possible [1, 3, 12, 27]. In other words, the assumptions in these works imply that the state space is already well-explored. Conversely, without such coverage (and, say, with sparse rewards), policy gradients often suffer from the vanishing gradient problem.

With regards to exploration, at least in the tabular setting, there is an established body of algorithms which provably explore in order to achieve sample efficient reinforcement learning, including model based methods [9, 13, 20, 29, 35, 36], model free approaches such as Q-learning [22, 31, 40, 60], Thompson sampling [5, 47, 51], and, more recently, policy optimization approaches [17, 24]. There are also a number of provable reinforcement learning algorithms balancing exploration and exploitation for MDPs with linearly parameterized dynamics, including [7, 17, 30, 32, 70–72].

The motivation for our work is to develop algorithms and guarantees which are more robust to violations in the underlying modeling assumptions. Indeed, the primary practical motivation for policy gradient methods is that the overall methodology is disentangled from modeling (and Markovian) assumptions, since they are an "end-to-end" approach, directly optimizing the cost function of interest. These practical considerations are supported by a body of theoretical results, both on direct policy optimization approaches [10, 34, 53, 54] and more recently on policy gradient approaches [3], which show that such incremental policy improvement approaches are amenable to function approximation and violations of modeling assumptions, under certain coverage assumptions over the state space.

This work focuses on how policy gradient methods can be extended to handle exploration, while also retaining their favorable properties with regards to how they handle function approximation and model misspecification. The practical relevance of answering these questions is evident by the growing body of empirical techniques for exploration which combine policy gradient methods with exploration bonuses such as pseudocounts [11], dynamics model errors [49], or random network distillation (RND) [16].

**Our Contributions.** This work introduces the POLICY COVER GUIDED POLICY GRADIENT algorithm (PC-PG), a direct, model-free, policy optimization approach which addresses exploration through the use of a learned ensemble of policies, the latter provides a policy cover over the state space. The use of a learned policy cover addresses exploration, and also addresses what is known as the "catastrophic forgetting" problem in policy gradient approaches (which use reward bonuses); while the on-policy nature avoids the "delusional bias" inherent to Bellman backup-based approaches, where approximation errors due to model misspecification amplify (see [42] for discussion).

It is a conceptually different approach from the predominant prior (and provable) RL algorithms, which are either model-based — variants of UCB [9, 13, 29, 36] or based on Thompson sampling [5, 51] — or model-free and value based, such as Q-learning [31, 60]. Our work adds policy optimization methods to this list, as a direct alternative: the use of learned covers permits a *a model-free approach* by allowing the algorithm to plan in the real world, using the cover for initializing the underlying policy optimizer. We remark that only a handful of prior (provable) exploration algorithms [31, 60] are model-free in the tabular setting, and these are largely value based.

At a high-level, we design an exploratory actor-critic method which performs critic estimation using linear functions in a given featurization $\phi(s, a)$ for a state $s$ and action $a$. Informally, our main result shows that for any policy $\pi$, we can compete with $V^\pi(s_0)$ in the initial state $s_0$, as long as we can estimate the critic functions for all actor policies discovered by the algorithm up to a small mean squared error, where the error is measured under the state-action distribution induced by the comparison policy $\pi$. While this distribution is unavailable to the algorithm, and the error cannot be explicitly reduced to zero by collecting more samples for fitting the critic, the assumption provides an expressivity condition on the features $\phi$. The condition is directly inspired by the recent characterization of *transfer error under distribution shift* as a key quantity underlying the convergence of policy optimization methods with function approximation by Agarwal et al. [3], and our results extend these ideas to the exploration setting.

This abstract result provides a string of striking corollaries for linear MDPs [32], state aggregation, and other settings, both when the modeling assumptions hold exactly and approximately. In the exact case, PC-PG is provably sample and computationally efficient for *both tabular and linear MDPs*. In the approximate case, the key upshot of our theory is to permit the use of certain average case error measures instead of $\ell_\infty$ conditions on misspecification in several prior works. Some of the key results that we establish include:

- **RKHS in Linear MDPs**: For the linear MDPs proposed by [32], our results hold when the linear MDP features live in an infinite dimensional Reproducing Kernel Hilbert Space (RKHS) (Theorem 4.1). It is not immediately evident how to extend the prior work on linear MDPs (e.g. [32]) to this setting (due to concentration issues with data re-use).

- **Bounded transfer error and state aggregation**: When specialized to a state aggregation setting, we show that PC-PG provides a different approximation guarantee in comparison to prior works. In particular, the aggregation need only be good locally, under the visitations of the comparison policy (Theorem 4.2). More generally, we analyze PC-PG under a notion of a small transfer error in critic fitting [3]—a condition on the error of a best on-policy critic under a comparison policy's

state distribution—which generalizes the special case of state aggregation, and show that PC-PG enjoys a favorable sample complexity whenever this transfer error is small (Theorem 4.3).

- **Empirical evaluation**: We provide experiments showing the viability of PC-PG in settings where prior bonus based approaches such as Random Network Distillation [16] do not recover optimal policies with high probability. Our experiments show our basic approach complements and leverages existing deep learning approaches, implicitly also verifying the robustness of PC-PG outside the regime where the sample complexity bounds provably hold.

## 2 Setting

A Markov Decision Process (MDP) $\mathcal{M} = (\mathcal{S}, \mathcal{A}, P, r, \gamma, s_0)$ is specified by a state space $\mathcal{S}$; an action space $\mathcal{A}$; a transition model $P : \mathcal{S} \times \mathcal{A} \to \Delta(\mathcal{S})$ (where $\Delta(\mathcal{S})$ denotes a distribution over states), a reward function $r : \mathcal{S} \times \mathcal{A} \to [0, 1]$, a discount factor $\gamma \in [0, 1)$, and a starting state $s_0$. We assume $\mathcal{A}$ is discrete and denote $A = |\mathcal{A}|$. Our results generalize to a starting state distribution $\mu_0 \in \Delta(\mathcal{S})$ but we use a single starting state $s_0$ to emphasize the need to perform exploration. A policy $\pi : \mathcal{S} \to \Delta(\mathcal{A})$ specifies a decision-making strategy in which the agent chooses actions based on the current state, i.e., $a \sim \pi(\cdot|s)$.

The value function $V^\pi(\cdot, r) : \mathcal{S} \to \mathbb{R}$ is defined as the expected discounted sum of future rewards, under reward function $r$, starting at state $s$ and executing $\pi$, i.e. $V^\pi(s; r) := \mathbb{E}\left[\sum_{t=0}^\infty \gamma^t r(s_t, a_t)|\pi, s_0 = s\right]$, where the expectation is taken with respect to the randomness of the policy and environment $\mathcal{M}$. The *state-action* value function $Q^\pi(\cdot, \cdot; r) : \mathcal{S} \times \mathcal{A} \to \mathbb{R}$ is defined as $Q^\pi(s, a; r) := \mathbb{E}\left[\sum_{t=0}^\infty \gamma^t r(s_t, a_t)|\pi, s_0 = s, a_0 = a\right]$.

We define the discounted state-action distribution $d_s^\pi$ of a policy $\pi$: $d_{s'}^\pi(s, a) := (1 - \gamma)\sum_{t=0}^\infty \gamma^t \mathrm{Pr}^\pi(s_t = s, a_t = a|s_0 = s')$, where $\mathrm{Pr}^\pi(s_t = s, a_t = a|s_0 = s')$ is the probability that $s_t = s$ and $a_t = a$, after we execute $\pi$ from $t = 0$ onwards starting at state $s'$ in model $\mathcal{M}$. Similarly, we define $d_{s',a'}^\pi(s, a)$ as: $d_{s',a'}^\pi(s, a) := (1 - \gamma)\sum_{t=0}^\infty \gamma^t \mathrm{Pr}^\pi(s_t = s, a_t = s|s_0 = s', a_0 = a')$. For any state-action distribution $\nu$, we write $d_\nu^\pi(s, a) := \sum_{(s',a') \in \mathcal{S} \times \mathcal{A}} \nu(s', a') d_{s',a'}^\pi(s, a)$. For ease of presentation, we assume that the agent can reset to $s_0$ at any point in the trajectory.[1] We denote $d_\nu^\pi(s) = \sum_a d_\nu^\pi(s, a)$. The goal of the agent is to find a policy $\pi$ that maximizes the expected value from the starting state $s_0$, i.e. the optimization problem is: $\max_\pi V^\pi(s_0)$, where the max is over some policy class. For completeness, we specify a $d_\nu^\pi$-sampler and an unbiased estimator of $Q^\pi(s, a; r)$ in Algorithm 1, which are standard in discounted MDPs. The $d_\nu^\pi$ sampler samples $(s, a)$ i.i.d from $d_\nu^\pi$, and the $Q^\pi$ sampler returns an unbiased estimate of $Q^\pi(s, a; r)$ for a given triple $(s, a, r)$ by a single roll-out from $(s, a)$.

We assume access to a feature map $\phi(s, a)$ for each $s \in \mathcal{S}$ and $a \in \mathcal{A}$, with $\|\phi(s, a)\| \le 1$, and $\|\cdot\|$ is a norm in some Reproducing Kernel Hilbert Space (RKHS). The features are primarily used for fitting the critic functions $Q^\pi$ in our algorithm, though some of our theoretical results also make stronger assumptions on the MDP dynamics in terms of $\phi$ (e.g., linear MDP).

**Notation.** When clear from context, we write $d^\pi(s, a)$ and $d^\pi(s)$ to denote $d_{s_0}^\pi(s, a)$ and $d_{s_0}^\pi(s)$ respectively, where $s_0$ is the starting state in our MDP. For iterative algorithms which obtain policies at each episode, we let $V^n, Q^n, A^n$ and $d^n$ denote the corresponding quantities associated with episode $n$. For a vector $v$, we denote $\|v\|_2 = \sqrt{\sum_i v_i^2}$, $\|v\|_1 = \sum_i |v_i|$, and $\|v\|_\infty = \max_i |v_i|$. For a matrix $V$, we define $\|V\|_2 = \sup_{x:\|x\|_2 \le 1} \|Vx\|_2$, and $\det(V)$ as the determinant of $V$. We use Uniform$(\mathcal{A})$ (in short Unif$_\mathcal{A}$) to represent a uniform distribution over the set $\mathcal{A}$.

## 3 The PC-PG Algorithm

To motivate the algorithm, first consider the original objective function: $\max_{\pi \in \Pi} V^\pi(s_0; r)$, where $r$ is the true reward function. Simply doing policy gradient ascent on this objective function may easily lead to poor stationary points due to lack of coverage (i.e. lack of exploration). In such cases, a more desirable objective function is of the form:

$$\text{A wide coverage objective:} \quad \max_{\pi \in \Pi} \mathbb{E}_{s_0, a_0 \sim \rho_{\text{cov}}} [Q^\pi(s_0, a_0; r)] \tag{1}$$

**Algorithm 1** $d^\pi$ sampler and $Q^\pi$ estimator

---

1: **function** $d_\nu^\pi$-SAMPLER
2:    **Input**: $\nu \in \Delta(\mathcal{S} \times \mathcal{A}), \pi, r(s,a)$
3:     Sample $s_0, a_0 \sim \nu$
4:     Execute $\pi$ from $s_0, a_0$; at any step $t$ with $(s_t, a_t)$, terminate the episode with probability $1 - \gamma$
5:    **Return**: $s_t, a_t$
6: **end function**
7: **function** $Q^\pi$-ESTIMATOR
8:    **Input**: current state-action $(s,a)$, reward $r(s,a)$, $\pi$
9:     Execute $\pi$ from $(s_0, a_0) = (s,a)$; at step $t$ with $(s_t, a_t)$, terminate with probability $1 - \gamma$
10:   **Return**: $\widehat{Q}^\pi(s,a) = \sum_{i=0}^{t} r(s_i, a_i)$ where $(s_0, a_0) = (s,a)$
11: **end function**

---

**Algorithm 2** POLICY COVER GUIDED POLICY GRADIENT (PC-PG)

---

1: **Input**: iterations $N$, threshold $\beta$, regularizer $\lambda$
2: Initialize $\pi^0(a|s)$ to be uniform
3: **for** episode $n = 0, \ldots N - 1$ **do**
4:    Estimate the covariance of $\pi^n$ as $\widehat{\Sigma}^n = \sum_{i=1}^{K} \phi(s_i, a_i)\phi(s_i, a_i)^\top / K$ with $\{s_i, a_i\}_{i=1}^{K} \sim d^n$
5:    Estimate the covariance of the policy cover as $\widehat{\Sigma}_{\text{cov}}^n := \sum_{i=0}^{n} \widehat{\Sigma}^i + \lambda I$
6:    Set the exploration bonus $b^n$ to reward infrequently visited state-action under $\rho_{\text{cov}}^n$ (3)

$$b^n(s,a) = \frac{\mathbf{1}\{(s,a) : \phi(s,a)^\top (\widehat{\Sigma}_{\text{cov}}^n)^{-1}\phi(s,a) \geq \beta\}}{1 - \gamma}. \qquad (2)$$

7:    Update $\pi^{n+1} = \text{NPG-Update}(\rho_{\text{cov}}^n, b^n)$ (Algorithm 3)
8: **end for**

---

where $\rho_{\text{cov}}$ is some initial state-action distribution which has wider coverage over the state space. As argued in [3, 34, 53, 54], wide coverage initial distributions $\rho_{\text{cov}}$ are critical to the success of policy optimization methods. However, in the RL setting, our agent can only start from $s_0$. Concretely, for the linear function approximation in terms of a given feature map $\phi$ that we employ here, from Theorem 6.1 of Agarwal et al. [3] (see also Abbasi-Yadkori et al. [1, 2]), we know that the coverage of $\rho_{\text{cov}}$ can be measured in terms of the smallest eigenvalue of its feature covariance matrix,[2] an object which plays a central role in our algorithm design.

The idea of our iterative algorithm, PC-PG (Algorithm 2), is to successively improve *both* the current policy $\pi$ and the coverage distribution $\rho_{\text{cov}}$. At episode $n$, we have $n + 1$ previous policies $\pi^0, \ldots \pi^n$. Each of these policies $\pi^i$ induces a distribution $d^i := d^{\pi^i}$ over the state-action space. Define $\rho_{\text{cov}}^n$ as:

$$\rho_{\text{cov}}^n(s,a) = \sum_{i=0}^{n} d^i(s,a)/(n+1) \qquad (3)$$

Intuitively, $\rho_{\text{cov}}^n$ reflects the coverage the algorithm has over the state-action space at the start of the $n$-th episode. PC-PG then uses $\rho_{\text{cov}}^n$ in the previous objective (1) with two modifications: PC-PG modifies the instantaneous reward function $r$ with a bonus $b^n$ in order to search for a policy $\pi^{n+1}$ which covers a novel part of space. It also modifies the policy class from $\Pi$ to $\Pi_{\text{bonus}}$, where all policies $\pi \in \Pi_{\text{bonus}}$ are constrained to simply take a random rewarding action for those states where the bonus is already large (see Eq 4 in Alg. 3). With this, PC-PG's objective at the $n$-th episode is: $\max_{\pi \in \Pi_{\text{bonus}}} \mathbb{E}_{s_0, a_0 \sim \rho_{\text{cov}}^n} [Q^\pi(s_0, a_0; r + b^n)]$. The idea is that PC-PG can effectively optimize over the region where $\rho_{\text{cov}}^n$ has coverage. Furthermore, by construction of the bonus, the algorithm is encouraged to escape the current region of coverage.

**Reward bonus construction.** At each episode $n$, PC-PG maintains an estimate of feature covariance of the policy cover $\rho_{\text{cov}}^n$ (Line 5 of Algorithm 2). Next we use this covariance matrix to identify

**Algorithm 3** Natural Policy Gradient (NPG) Update

---

1: **Input** $\rho_{\text{cov}}^n$, $b^n$, learning rate $\eta$, sample size $M$ for critic fitting, iterations $T$
2: Define $\mathcal{K}^n = \{s : \forall a \in \mathcal{A}, b^n(s,a) = 0\}$
3: Initialize policy $\pi^0 : \mathcal{S} \to \Delta(\mathcal{A})$, such that

$$\pi^0(\cdot|s) = \begin{cases} \text{Uniform}(\mathcal{A}) & s \in \mathcal{K}^n \\ \text{Uniform}(\{a \in \mathcal{A} : b^n(s,a) > 0\}) & s \notin \mathcal{K}^n. \end{cases}$$

4: **for** $t = 0 \to T - 1$ **do**
5:   Draw $M$ i.i.d samples $\left\{s_i, a_i, \widehat{Q}^{\pi^t}(s_i, a_i; r + b^n)\right\}_{i=1}^M$ with $s_i, a_i \sim \rho_{\text{cov}}^n$ (see Alg 1)
6:   **Critic** fit:

$$\theta^t = \underset{\|\theta\| \leq W}{\text{argmin}} \sum_{i=1}^M \left(\theta \cdot \phi(s_i, a_i) - \left(\widehat{Q}^{\pi^t}(s_i, a_i; r + b^n) - b^n(s_i, a_i)\right)\right)^2$$

7:   **Actor** update

$$\pi^{t+1}(\cdot|s) \propto \pi^t(\cdot|s) \exp\left(\eta\left(b^n(s, \cdot) + \theta^t \cdot \phi(s, \cdot)\right) \mathbf{1}\{s \in \mathcal{K}^n\}\right) \tag{4}$$

8: **end for**
9: **return** $\pi := \text{argmax}_{\pi \in \{\pi^0, \dots, \pi^{T-1}\}} V^\pi(s_0; r + b^n)$

---

state-action pairs which are adequately covered by $\rho_{\text{cov}}^n$, inspired by prior policy optimization results for linear function approximation as mentioned before. The goal of the reward bonus is to identify state, action pairs whose features are less explored by $\rho_{\text{cov}}^n$ and incentivize visiting them. The bonus $b^n(s,a)$ defined in Eq 2 achieves this. If $\widehat{\Sigma}_{\text{cov}}^n$ has a small eigenvalue along $\phi(s,a)$, then we assign the largest possible reward-to-go (i.e., $1/(1-\gamma)$) for this $(s,a)$ pair to encourage exploration.[3] Note that reward bonus is only explicitly computed on states along the trajectory roll-outs during the execution of the algorithm (see Algorithm 3).

**Policy Optimization.**    With the bonus, we update the policy via $T$ steps of natural policy gradient (Algorithm 3). In the NPG update, we first approximate the value function $Q^{\pi^t}(s, a; r + b^n)$ under the policy cover $\rho_{\text{cov}}^n$ (line 6). Specifically, we use linear function approximator to approximate $Q^{\pi^t}(s, a; r + b^n) - b^n(s, a)$ via constrained linear regression (line 6), and then approximate $Q^{\pi^t}(s, a; r + b^n)$ by adding bonus back: $\overline{Q}_{b^n}^t(s, a) := b^n(s, a) + \theta^t \cdot \phi(s, a)$, Note that the error of $\overline{Q}_{b^n}^t(s, a)$ to $Q^{\pi^t}(s, a; r + b^n)$ is simply the prediction error of $\theta^t \cdot \phi(s, a)$ to the regression target $Q^{\pi^t}(s, a; r + b^n) - b^n$. The purpose of structuring the value function estimation this way, instead of directly approximating $Q^t(s, a; r + b^n)$ with a linear function, for instance, is that the regression problem defined in line 6 will have a good linear solution for the special case linear MDPs, while we cannot guarantee the same for $Q^t(s, a; r + b^n)$ due to the non-linearity of the bonus. We then use the critic $\overline{Q}_{b^n}^t$ for updating policy (Eq. (4)). These are the exponential gradient updates (as in [3, 33]), but are constrained for $s \in \mathcal{K}^n$ (see line 2 for the definition of $\mathcal{K}^n$). The initialization and the update ensure that $\pi^t$ chooses actions uniformly from $\{a : b^n(s, a) > 0\} \subseteq \mathcal{A}$ at any state $s$ with $|\{a : b^n(s, a) > 0\}| > 0$ (the policy is restricted to act uniformly among positive bonus actions). Our policy update ensures that for any state $s$, $\pi^{t+1}$ is not that different from $\pi^t$. Similar to CPI [34] and PSDP [10], such incremental policy update is the key to the robustness in model-misspecification that goes beyond the worst case $\ell_\infty$ assumption, which we discuss in detail in Sec. 4.

## 4   Theory

We first state sample complexity results for linear MDPs. We then demonstrate the robustness of PC-PG to model misspecification. We show that in state aggregation, error incurred is only an average model error from aggregation averaged over the fixed comparator's abstracted state distribution.

For more general agnostic setting we show that our algorithm is robust to model-misspecification which is measured in a new concept of *transfer error* introduced by [3] recently. In Appendix G, we further provide model-misspecification examples and show why most prior approaches fail due to the delusional bias of Bellman backups under function approximation and model misspecification [42].

## 4.1 Well specified case: Linear MDPs in RKHS

We directly work on linear MDPs in a general Reproducing Kernel Hilbert space (RKHS).

**Definition 4.1** (Linear MDP in RKHS). *Let $\mathcal{H}$ be an RKHS, and define a feature mapping $\phi$ : $\mathcal{S} \times \mathcal{A} \to \mathcal{H}$. An MDP $(\mathcal{S}, \mathcal{A}, P, r, \gamma, s_0)$ is called a linear MDP if the reward function lives in $\mathcal{H}$: $r(s, a) = \langle \theta, \phi(s, a) \rangle_{\mathcal{H}}$, and the transition operator $P(s'|s, a)$ also lives in $\mathcal{H}$: $P(s'|s, a) = \langle \mu(s'), \phi(s, a) \rangle_{\mathcal{H}}$ for all $(s, a, s')$. Denote $\mu$ as a matrix whose each row corresponds to $\mu(s)$. We assume the parameter norms[4] are bounded as $\|\theta\| \leq \omega$, $\|v^\top \mu\| \leq \xi$ for all $v \in \mathbb{R}^{|\mathcal{S}|}$ with $\|v\|_\infty \leq 1$, $\sup_{s,a} \|\phi(s, a)\| \leq 1$. we assume that the initial state $s_0$ has lower bounded non-zero norm $\min_{a \in \mathcal{A}} \|\phi(s_0, a)\| \geq c_0 \in (0, 1]$ (other states could have arbitrarily small norms).*

As our feature vector $\phi$ could be infinite dimensional, to measure the sample complexity, we define the *maximum information gain* of the underlying MDP $\mathcal{M}$. First, denote the covariance matrix of any policy $\pi$ as $\Sigma^\pi = \mathbb{E}_{(s,a) \sim d^\pi} \left[ \phi(s, a) \phi(s, a)^\top \right]$. We define the maximum information gain below:

**Definition 4.2** (Maximum Information Gain $\mathcal{I}_N(\lambda)$). *We define the maximum information gain as:* $\mathcal{I}_N(\lambda) := \max_{\{\pi^i\}_{i=0}^{N-1}} \log \det \left( \frac{1}{\lambda} \sum_{i=0}^{N-1} \Sigma^{\pi^i} + I \right)$, *where $\lambda \in \mathbb{R}^+$.*

**Remark 4.1.** This quantity is identical to the maximum information gain in Gaussian Process bandits [59] from a Bayesian perspective. A related quantity occurs in a more restricted linear MDP model [69]. Note that when $\phi(s, a) \in \mathbb{R}^d$, we have that $\log \det \left( \sum_{i=1}^{n} \Sigma^{\pi^i}/\lambda + I \right) \leq d \log(nB^2/\lambda + 1)$ assuming $\|\phi(s, a)\|_2 \leq B$, which means that the information gain is always at most $\widetilde{O}(d)$.

**Theorem 4.1** (Sample Complexity of PC-PG for Linear MDPs). *Fix $\epsilon, \delta \in (0, 1)$ and an arbitrary comparator policy $\pi^\star$ (not necessarily an optimal policy). Suppose that $\mathcal{M}$ is a linear MDP (4.1). There exists a setting of the parameters such that PC-PG uses a number of samples at most $poly \left( \frac{1}{1-\gamma}, \log(A), \frac{1}{\epsilon}, \mathcal{I}_N(1), \omega, \xi, \ln \left( \frac{1}{\delta} \right) \right)$ and, with probability greater than $1 - \delta$, returns a policy $\widehat{\pi}$ such that $V^{\widehat{\pi}}(s_0) \geq V^{\pi^\star}(s_0) - \epsilon$.*

A few remarks are in order:

**Remark 4.2.** For tabular MDPs, as $\phi$ is a $|\mathcal{S}||\mathcal{A}|$ indictor vector, the theorem above immediately extends to tabular MDPs with $\mathcal{I}_N(1)$ being replaced by $|\mathcal{S}||\mathcal{A}| \log(N + 1)$.

**Remark 4.3.** In contrast with LSVI-UCB [32], PC-PG works for infinite dimensional $\phi$ with a polynomial dependency on the maximum information gain $\mathcal{I}_N(1)$.

Instead of proving Theorem 4.1 directly, we will state and prove a general theorem of PC-PG for general MDPs with model-misspecification measured in a new concept *transfer error* (Assumption 4.1) introduced by [3] in Section 4.3. Theorem 4.1 can be understood as a corollary of a more general agnostic theorem (Theorem 4.3). Detailed proof of Theorem 4.1 is included in Appendix E.

## 4.2 State-Aggregation under Model Misspecification

Consider a simple model-misspecified setting where the model error is introduced due to state action aggregation. Suppose we have an aggregation function $\phi : \mathcal{S} \times \mathcal{A} \to \mathcal{Z}$, where $\mathcal{Z}$ is a finite categorical set, the "state abstractions", which we typically think of as being much smaller than the (possibly infinite) number of state-action pairs. Intuitively, we aggregate state-action pairs that have similar transitions and rewards to an abstracted state $z$. This aggregation introduces model-misspecification.

**Definition 4.3.** *We define model-misspecification $\epsilon_z$ for any $z \in \mathcal{Z}$ as*

$$\epsilon_z := \max_{(s,a),(s',a') \text{ s.t. } \phi(s,a) = \phi(s',a') = z} \left\{ \|P(\cdot|s, a) - P(\cdot|s', a')\|_1, |r(s, a) - r(s', a')| \right\}.$$

The model-misspecification measures the maximum possible disagreement in terms of transition and rewards of two state-action pairs which are mapped to the same abstracted state.

The folklore result is that with the definition $\|\epsilon_{misspec}\|_\infty := \max_{z \in \mathcal{Z}} \epsilon_z$, algorithms such as UCB and $Q$-learning, and regular policy gradient [52] succeed with an additional additive error of $\|\epsilon_{misspec}\|_\infty / (1 - \gamma)^2$, and will have sample complexity guarantees that are polynomial in only $|\mathcal{Z}|$. Interestingly, see [22, 40, 52] for conditions limited to only $Q^\star$ are still *global* in nature. The following theorem shows PC-PG only requires a more local guarantee where our aggregation needs to be only good under the distribution of abstracted states where an optimal policy tends to visit.

**Theorem 4.2** (Misspecified, State-Aggregation Bound). *Fix $\epsilon, \delta \in (0, 1)$. Let $\pi^\star$ be an arbitrary comparator policy. There exists a setting of the parameters such that PC-PG (Algorithm 2) uses a total number of samples at most poly $\left( |\mathcal{Z}|, \log(A), \frac{1}{1-\gamma}, \frac{1}{\epsilon}, \ln\left(\frac{1}{\delta}\right) \right)$ and, with probability greater than $1 - \delta$, returns a policy $\widehat{\pi}$ such that, $V^{\widehat{\pi}}(s_0) \geq V^{\pi^\star}(s_0) - \epsilon - \frac{2\mathbb{E}_{s \sim d^{\pi^\star}} \max_a \left[ \epsilon_{\phi(s,a)} \right]}{(1-\gamma)^3}$.*

Here, it could be that $\mathbb{E}_{s \sim d^{\pi^\star}} \max_a [\epsilon_{\phi(s,a)}] \ll \|\epsilon_{misspec}\|_\infty$ due to that our error notion is an average case one under the comparator. We refer readers to Appendix F for detailed proof of the above theorem which can also be regarded as a corollary of a more general agnostic theorem (Theorem 4.3) that we present in the next section. Note that here we pay an additional $1/(1 - \gamma)$ factor in the approximation error due to the fact that after reward bonus, we have $r(s, a) + b^n(s, a) \in [0, 1/(1-\gamma)]$. [5] One point worth reflecting on is how few guarantees there are in the more general RL setting (beyond dynamic programming), which address model-misspecification beyond global $\ell_\infty$ bounds. Our conjecture is that this is not merely an analysis issue but an algorithmic one, where incremental algorithms such as PC-PG are required for strong misspecified algorithmic guarantees. We return to this point in Appendix G, with an example showing why this might be the case.

### 4.3 Agnostic Guarantees with Bounded Transfer Error

We now consider a general MDP in this section, where we do not assume the linear MDP modeling assumptions hold. As $Q - b^n$ may not be linear with respect to the given feature $\phi$, we need to consider model misspecification due to the linear function approximation with features $\phi$. We use the new concept of transfer error from [3] below. We use the shorthand notation: $Q_{b^n}^t(s, a) = Q^{\pi^t}(s, a; r + b^n)$ below. We capture model misspecification using the following assumption.

**Assumption 4.1** (Bounded Transfer Error). *With respect to a target function $f : \mathcal{S} \times \mathcal{A} \to \mathbb{R}$, define the critic loss function $L(\theta; d, f)$ with $d \in \Delta(\mathcal{S} \times \mathcal{A})$ as: $L(\theta; d, f) := \mathbb{E}_{(s,a) \sim d} (\theta \cdot \phi(s, a) - f)^2$, which is the square loss of using the critic $\theta \cdot \phi$ to predict a given target function $f$, under distribution $d$. Consider an arbitrary comparator policy $\pi^\star$ (not necessarily an optimal policy) and denote the state-action distribution $d^\star(s, a) := d^{\pi^\star}(s) \circ Unif_\mathcal{A}(a)$. For all episode $n$ and all iteration $t$ inside episode $n$, define: $\theta_\star^t \in \operatorname{argmin}_{\|\theta\| \leq W} L(\theta; \rho_{cov}^n, Q_{b^n}^t - b^n)$. Then we assume that (when running Algorithm 2), $\theta_\star^t$ has a bounded prediction error when transferred to $d^\star$ from $\rho_{cov}^n$; more formally:*

$$L\left(\theta_\star^t; d^\star, Q_{b^n}^t - b^n\right) \leq \epsilon_{bias} \in \mathbb{R}^+.$$

Note that the transfer error $\epsilon_{bias}$ measures the prediction error, at episode $n$ and iteration $t$, of a best on-policy fit $\overline{Q}_{b^n}^t(s, a) := b^n(s, a) + \theta_\star^t \cdot \phi(s, a)$ measured under a fixed distribution $d^\star$ from the fixed comparator (note $d^\star$ is different from the training distribution $\rho_{cov}^n$ hence the name *transfer*).

This assumption first appears in the recent work of Agarwal et al. [3] in order to analyze NPG under linear function approximation. This is a milder notion of model misspecification than $\ell_\infty$-variants more prevalent in the literature, as it is an average-case quantity which can be significantly smaller in favorable cases. With the above assumption on the transfer error, the next theorem states an agnostic result for the sample complexity of PC-PG:

**Theorem 4.3** (Agnostic Guarantee of PC-PG). *Fix $\epsilon, \delta \in (0, 1)$ and consider an arbitrary comparator policy $\pi^\star$ (not necessarily an optimal policy). Assume Assumption 4.1 holds, and $\sup_{s,a} \|\phi(s, a)\| \leq 1$, and $\sup_a \|\phi(s_0), a\| \geq c_0 \in (0, 1]$. There exists a setting of the parameters $(\beta, \lambda, K, M, \eta, N, T)$*

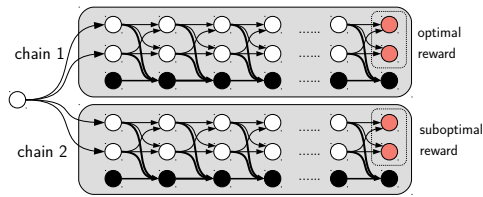

| Algorithm | Horizon | | | |
|---|---|---|---|---|
| | 2 | 5 | 10 | 15 |
| PPO | 1.0 | 0.0 | 0.0 | 0.0 |
| PPO+RND | 0.75 | 0.40 | 0.50 | 0.55 |
| PC-PG | 1.0 | 1.0 | 1.0 | 1.0 |

Figure 1: **Left** panel shows the Bidirectional Diabolical Combination Lock domain (see text for details). **Right** panel shows success rate of different algorithms averaged over 20 different seeds.

such that PC-PG uses a number of samples at most poly $\left( \frac{1}{1-\gamma}, \log(A), \frac{1}{\epsilon}, \mathcal{I}_N(1), W, \ln\left(\frac{1}{\delta}\right) \right)$ and, with probability greater than $1 - \delta$, returns a policy $\widehat{\pi}$ such that $V^{\widehat{\pi}}(s_0) \geq V^{\pi^\star}(s_0) - \epsilon - \frac{\sqrt{2A\epsilon_{bias}}}{1-\gamma}$.

The precise polynomial of the sample complexity, along with the settings of all the hyperparameters — $\beta$ (threshold for bonus), $\lambda$, $K$ (samples for estimating cover's covariance), $M$ (samples for fitting critic), $\eta$ (learning rate in NPG), $N$ (number of episodes), and $T$ (number of NPG iterations per episode) — is provided in Theorem D.1 (Appendix D), where we discuss two examples of $\phi$—finite dimensional $\phi \in \mathbb{R}^d$, and infinite dimensional $\phi$ in RKHS with RBF kernel (Remark D.1).

For well-specified cases such as tabular MDPs and linear MDPs, due to $Q^\pi(\cdot, \cdot; r + b^n) - b^n$ is always a linear function with respect to the features, one can easily show that $\epsilon_{bias} = 0$ (which we show in Appendix E), as one can pick the best on-policy fit $\theta^t_\star$ to be the exact linear representation of $Q^\pi(s, a; r + b^n) - b^n(s, a)$. Further, in the state-aggregation example, we can show that $\epsilon_{bias}$ is upper bounded by the expected model-misspecification with respect to the comparator policy's distribution (Appendix F). We refer readers to [3] for detailed discussion with respect to the comparison of transfer error and usual concentrability assumptions [3, 34, 53].

# 5 Experiments

We provide experiments illustrating PC-PG's performance on problems requiring exploration, and focus on showing the algorithm's flexibility to leverage existing policy gradient algorithms with neural networks (e.g., PPO [56]). Specifically, we show that for challenging exploration tasks, our algorithm combined with PPO significantly outperforms both vanilla PPO as well as PPO augmented with the popular RND exploration bonus [16]. For all experiments, we use policies parameterized by fully-connected or convolutional neural networks. We use a kernel $\phi(s, a)$ to compute bonus as $b(s, a) = \phi(s, a)^\top \hat{\Sigma}_{cov}^{-1} \phi(s, a)$, where $\hat{\Sigma}_{cov}$ is the empirical covariance matrix of the policy cover. In order to prune any redundant policies from the cover, we use a rebalancing scheme to select a policy cover which induces maximal coverage over the state space. This is done by finding weights $\alpha^{(n)} = (\alpha_1^{(n)}, ..., \alpha_n^{(n)})$ on the simplex at each episode which solve the optimization problem: $\alpha^{(n)} = \text{argmax}_\alpha \log \det \left[ \sum_{i=1}^n \alpha_i \hat{\Sigma}_i \right]$ where $\hat{\Sigma}_i$ is the empirical covariance matrix of $\pi_i$. Details of the implemented algorithm, network architectures and kernels can be found in Appendix I.

**Bidirectional Diabolical Combination Lock** We first provide experiments on an exploration problem designed to be particularly difficult: the Bidirectional Diabolical Combination Lock (a harder version of the problem in [43], see Figure 1). In this problem, the agent starts at an initial state $s_0$ (left most state), and based on its first action, transitions to one of two combination locks of length $H$. Each combination lock consists of a chain of length $H$, at the end of which are two states with high reward. At each level in the chain, 9 out of 10 actions lead the agent to a dead state (black) from which it cannot recover and lead to zero reward. The problem is challenging for exploration for several reasons: (1) *Sparse positive rewards*: Uniform exploration has a $10^{-H}$ chance of reaching a high reward state; (2) *Dense antishaped rewards*: The agent receives a reward of $-1/H$ for transitioning to a good state and $0$ to a dead state. A locally optimal policy is to transition to a dead state quickly; (3) *Forgetting*: At the end of one of the locks, the agent receives a maximal reward of $+5$, and at the end of the other lock it receives a reward of $+2$. Since there is no indication which lock has the optimal reward, agent needs to remember to visit both chains. For both the policy network input and the kernel we used a binary vector encoding the current lock, state and time step

| Policy 1 | Policy 4 | Policy 5 | Policy 11 | Policy 12 |

Agent
start location ⟶

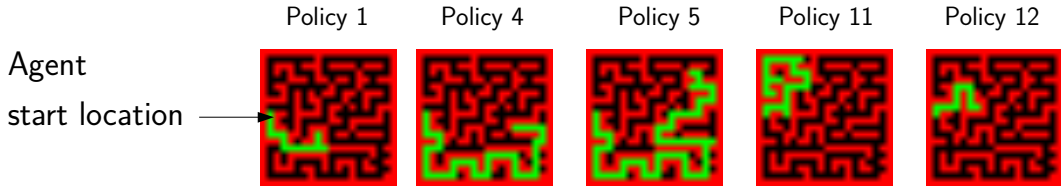

Figure 2: Different policies' trajectory (green) in the policy cover for the maze environment.

as one-hot components. We compared to two other methods: a PPO agent, and a PPO agent with a RND exploration bonus, all of which used the same representation as input.

Figure 1 shows the PPO agent succeeds for the shortest problem of horizon $H = 2$. The PPO+RND agent succeeds roughly $50\%$ of the time: it avoids the local minimum and explores to the end of one of the chains. However the agent's policy quickly becomes deterministic and the agent forgets to go back and explore the other chain after it has reached the reward at the end of the first (as shown in Figure 5 (a)), PC-PG succeeds over all seeds and horizon lengths. We found that the policy cover provides near uniform coverage over both chains (see Figure 5 (b)).

**Reward-free Exploration in Mazes** We evaluated PC-PG in a reward-free setting using maze environments adapted from [46]. The agent's observation consists of an RGB-image of the maze with the red channel representing the walls and the green channel representing the location of the agent (an example is shown in Figure 2). We compare PC-PG, PPO and PPO+RND where the agent receives a constant environment reward of 0

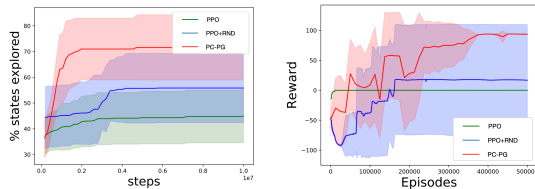

Figure 3: Results for maze (left) & control (right).

(note that PPO receives zero gradient; PC-PG and PPO+RND learn from their reward bonus). Figure 3 (left) shows the percentage of locations in the maze visited by each of the agents over the course of 10 million steps. The proportion of states visited by the PPO agent stays relatively constant, while the PPO+RND agent is able to explore to some degree. PC-PG quickly visits a significantly higher proportion of locations. Visualizations of traces from policies can be seen in Figure 2.

**Continuous Control** We further evaluated PC-PG on continuous control MountainCar from OpenAI Gym [15]. Note here actions are continuous in $[-1, 1]$ and incur a small negative reward. Since the agent only receives a large reward $(+100)$ if it reaches the top of the hill, a locally optimal policy is to do nothing (e.g., PPO never escapes this local optimality in our experiments). Results for PPO, PPO+RND and PC-PG are shown in Figure 3 (right). The PPO agent quickly learns the locally optimal policy. The PPO+RND agent exhibits wide variability across seeds: some seeds solve the task while others not. The PC-PG agent consistently discovers a good policy across all seeds. In Figure 6, we show the traces of policies in the policy cover constructed by PC-PG.

## 6 Discussion

This work proposes a new policy gradient algorithm for balancing the exploration-exploitation tradeoff in RL, which enjoys provable sample efficiency guarantees in the linear and kernelized settings. Our experiments provide evidence that the algorithm can be combined with neural policy optimization methods and be effective in practice. An interesting direction for future work would be to combine our approach with feature learning [4]. in rich observation settings to learn a good feature representation.

## Broader Impact

This paper provides a provably efficient policy gradient algorithm. Though the nature of the paper is mostly theoretical and the paper heavily focuses on understanding the theoretical foundations of one of the most popular RL algorithms, i.e., policy gradient, we believe that our theoretical findings and the proposed new algorithm will have a broader impact on the society. Due to the high sample complexity of existing PG methods, they are often limited to applications related to video games. We believe that by providing global optimality and sample efficiency to PG methods, we will significantly broaden the application scope of PG methods. Specifically, our work potentially could enable PG methods to be deployed in real-world applications such as precision medicine, human robot interaction, and personalized eduction systems where safety and robustness are critical.

## Acknowledgments and Disclosure of Funding

Part of the work was done when MH and WS were at Microsoft Research NYC. The authors would like to thank Andrea Zanette, Ching-An Cheng, and Xuezhou Zhang for carefully reviewing the proofs, and Akshay Krishnamurthy for helpful discussions. Sham Kakade gratefully acknowledges funding from the ONR award N00014-18-1-2247, and NSF Awards CCF-1703574 and CCF-1740551.

## Footnotes

[1]This can be replaced with a termination at each step with probability $1 - \gamma$.

[2]For infinite dimensions, the more general notion of relative condition number is required, but minimum eigenvalue provides reasonable intuition and is sufficient in the finite dimensional case.

[3] For an infinite dimensional RKHS, the bonus can be computed in the dual using the kernel trick (e.g., [66]).

[4]The norms are induced by the inner product in the Hilbert space $\mathcal{H}$, unless stated otherwise.

[5]We note that instead of using reward bonus, we could construct absorbing MDPs to make rewards scale $[0, 1]$. This way we will pay $1/(1 - \gamma)^2$ in the approximation error instead.

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
