[Supplementary Material]

# A   Additional Related Work

In this section, we discuss additional related works.

We first discuss work with regards to policy gradient methods and incremental policy optimization; we then discuss work with regards to exploration in the context of explicit (or implicit) assumptions on the MDP (which permit sample complexity that does not explicitly depend on the number of states); and then "on-policy" exploration methods. Finally, we discuss the recent and concurrent work of Cai et al. [17], Efroni et al. [24], which provide an optimistic policy optimization approach which uses off-policy data.

Our line of work seeks to extend the recent line of provably correct policy gradient methods [1, 3, 8, 12, 25, 26, 41, 45] to incorporate exploration. As discussed in the intro, our focus is that policy gradient methods, and more broadly "incremental" methods — those methods which make gradual policy changes such as Conservative Policy Iteration (CPI) [34, 53, 54], Policy Search by Dynamic Programming (PSDP) [10], and MD-MPI [27] — have guarantees with function approximation that are stronger than the more abrupt approximate dynamic programming methods, which rely on the boundedness of the more stringent concentrability coefficients [6, 44, 63]; see Agarwal et al. [3], Chen and Jiang [18], Geist et al. [27], Scherrer [53], Shani et al. [58] for further discussion. Our main agnostic result shows how PC-PG is more robust than all extant bounds with function approximation in terms of both concentrability coefficients and distribution mismatch coefficients; as such, our results require substantially weaker assumptions, building on the recent work of Agarwal et al. [3] who develop a similar notion of robustness in the policy optimization setting without exploration. Specifically, when specializing to linear MDPs and tabular MDPs, our algorithm is PAC while algorithms such as CPI and NPG are not PAC without further assumption on the reset distribution [3].

We now discuss results with regards to exploration in the context of explicit (or implicit) assumptions on the underlying MDP. To our knowledge, all prior works only provide provable algorithms, under either realizability assumptions or under well specified modelling assumptions; the violations tolerated in these settings are, at best, in an $\ell_\infty$-bounded, worst case sense. The most general set of results are those in [30], which proposed the concept of Bellman Rank to characterize the sample complexity of value-based learning methods and gave an algorithm that has polynomial sample complexity in terms of the Bellman Rank, though the proposed algorithm is not computationally efficient. Bellman rank is bounded for a wide range of problems, including MDPs with small number of hidden states, linear MDPs, LQRs, etc. Later work gave computationally efficient algorithms for certain special cases [21, 23, 32, 43, 69]. Recently, Witness rank, a generalization of Bellman rank to model-based methods, was proposed by [61] and was later extended to model-based reward-free exploration by [28]. We focus on the linear MDP model, studied in [32, 69]. We note that Yang and Wang [69] also prove a result for a type of linear MDPs, though their model is significantly more restrictive than the model in  Jin et al. [32]. Another notable result is due to Wen and Van Roy [67], who showed that in deterministic systems, if the optimal $Q$-function is within a pre-specified function class which has bounded Eluder dimension (for which the class of linear functions is a special case), then the agent can learn the optimal policy using a polynomial number of samples; this result has been generalized by [23] to deal with stochastic rewards, using further assumptions such as low variance transitions and strictly positive optimality gap.

With regards to "on-policy" exploration methods, to our knowledge, there are relatively few provable results which are limited to the tabular case.  These are all based on Q-learning with uncertainty bonuses in the tabular setting, including the works in  [31, 60]. More generally, there are a host of results in the tabular MDP setting that handle exploration, which are either model-based or which re-use data (the re-use of data is often simply planning in the empirical model), which include [5, 9, 14, 20, 29, 35, 36, 38, 39, 64].

Cai et al. [17], Efroni et al. [24] recently study algorithms based on exponential gradient updates for tabular MDPs, utilizing the mirror descent analysis first developed in  [25] along with idea of optimism in the face of uncertainty. Both approaches use a critic computed from off-policy data and can be viewed as model-based, since the algorithm stores all previous off-policy data and plans in what is effectively the empirically estimated model (with appropriately chosen uncertainty bonuses); in constrast, the model-free approaches such as $Q$-learning do not store the empirical model and have a substantially lower memory footprint (see [31] for discussion on this latter point). Cai et al. [17]

further analyze their algorithm in the linear kernel MDP model [72], which is a different model from what is referred to as the linear MDP model Jin et al. [32]. Notably, neither model is a special case of the other. It is worth observing that the linear kernel MDP model of [72] is characterized by at most $d$ parameters, where $d$ is the feature dimensionality, so that model-based learning is feasible; in contrast, the linear MDP model of Jin et al. [32] requires a number of parameter that is $S \cdot d$ and so it is not describable using a small number of parameters (and yet, sample efficient RL is still possible). See Jin et al. [32] for further discussion.

# B  NPG Analysis (Algorithm 3)

In this section, we analyze Algorithm 3 for a particular episode $n$.

In order to carry out our analysis, we first set up some auxiliary MDPs which are needed in our analysis. Throughout this section, we focus on episode $n$.

## B.1  Set up of Augmented MDPs

Denote

$$\mathcal{K}^n := \left\{ (s,a) : \phi(s,a)^\top \left( \Sigma_{\text{cov}}^n \right)^{-1} \phi(s,a) \le \beta \right\}. \tag{5}$$

That is, $\mathcal{K}^n$ contains state-action pairs that obtain no reward bonuses. We abuse notation a bit by denoting $s \in \mathcal{K}^n$ if and only if $(s,a) \in \mathcal{K}^n$ for all $a \in \mathcal{A}$.

We also add an extra action denoted as $a^\dagger$ in $\mathcal{M}^n$. For any $s \notin \mathcal{K}^n$, we add $a^\dagger$ to the set of available actions one could take at $s$. We set rewards and transitions as follows:

$$r^n(s,a) = r(s,a) + b^n(s,a) + \mathbf{1}\{a = a^\dagger\}; \quad P^n(\cdot|s,a) = P(\cdot|s,a), \forall (s,a), \quad P^n(s|s,a^\dagger) = 1, \tag{6}$$

where $r(s,a^\dagger) = b^n(s,a^\dagger) = 0$ for any $s$.

Note that at this point, we have three different kinds of MDPs that we will cross during the analysis:

1. the original MDP $\mathcal{M}$—the one that PC-PG is ultimately optimizing;
2. the MDP with reward bonus $b^n(s,a)$—the one is optimized by NPG in each episode $n$ *in the algorithm*, which we denote as $\mathcal{M}_{b^n} = \{P, r(s,a) + b^n(s,a)\}$ with $P$ and $r$ being the transition and reward from $\mathcal{M}$;
3. the MDP $\mathcal{M}^n$ that is constructed in Eq. (6) which is *only used in analysis but not in algorithm*.

The relationship between $\mathcal{M}_{b^n}$ (item 2) and $\mathcal{M}^n$ (item 3) is that NPG Algorithm 3 runs on $\mathcal{M}_{b^n}$ (NPG is not even aware of the existence of $\mathcal{M}^n$) but we use $\mathcal{M}^n$ to analyze the performance of NPG below.

**Additional Notations.**  We are going to focus on a fixed comparator policy $\widetilde{\pi} \in \Pi$. We denote $\widetilde{\pi}^n$ as the policy such that $\widetilde{\pi}(\cdot|s) = \widetilde{\pi}^n(\cdot|s)$ for $s \in \mathcal{K}^n$, and $\widetilde{\pi}^n(a^\dagger|s) = 1$ for $s \notin \mathcal{K}^n$. This means that the comparator policy $\widetilde{\pi}^n$ will self-loop in a state $s \notin \mathcal{K}^n$ and collect maximum rewards. We denote $\widetilde{d}_{\mathcal{M}_n}$ as the state-action distribution of $\widetilde{\pi}^n$ under $\mathcal{M}^n$, and $V_{\mathcal{M}^n}^\pi, Q_{\mathcal{M}^n}^\pi$, and $A_{\mathcal{M}^n}^\pi$ as the value, Q, and advantage functions of $\pi$ under $\mathcal{M}^n$. We also use $Q_{b^n}^\pi(s,a)$ in as a shorthand for $Q^\pi(s,a;r+b^n)$, similarly $A_{b^n}^\pi(s,a)$ for $A^\pi(s,a;r+b^n)$, and $V_{b^n}^\pi(s)$ for $V^\pi(s;r+b^n)$.

**Remark B.1.** Note that policies used in the algorithm do not pick $a^\dagger$ (i.e., algorithms does not even aware of $\mathcal{M}^n$). Hence for any policy $\pi$ that we would encounter during learning, we have $V_{\mathcal{M}^n}^\pi(s) = V_{b^n}^\pi(s)$ for all $s$, $Q_{\mathcal{M}^n}^\pi(s,a) = Q_{b^n}^\pi(s,a)$ and $A_{\mathcal{M}^n}^\pi(s,a) = A_{b^n}^\pi(s,a)$ for all $s$ with $a \ne a^\dagger$. This fact is important as our algorithm is running on $\mathcal{M}_{b^n}$ while the performance progress of the algorithm is tracked under $\mathcal{M}^n$.

## B.2  Performance of NPG (Algorithm 3) on the Augmented MDP $\mathcal{M}^n$

In this section, we focus on analyzing the performance of NPG (Algorithm 3) on a specific episode $n$. Specifically we leverage the Mirror Descent analysis similar to Agarwal et al. [3] to show that regret

between the sequence of learned policies $\{\pi^t\}_{t=1}^T$ and the comparator $\widetilde{\pi}^n$ on the constructed MDP $\mathcal{M}^n$.

Via performance difference lemma [35], we immediately have:

$$V_{\mathcal{M}^n}^{\widetilde{\pi}^n} - V_{\mathcal{M}^n}^{\pi} = \frac{1}{1-\gamma} \mathbb{E}_{(s,a)\sim\widetilde{d}_{\mathcal{M}^n}} \left[A_{\mathcal{M}^n}^{\pi}(s,a)\right].$$

For notation simplicity below, given a policy $\pi$ and state $s$, we use $\pi_s$ to abbreviate $\pi(\cdot|s)$.

**Lemma B.1** (NPG Convergence). *Consider any episode $n$. Setting $\eta = \sqrt{\frac{\log(A)}{W^2 T}}$, assume NPG updates policy as:*

$$\pi^{t+1}(\cdot|s) \propto \begin{cases} \pi^t(\cdot|s)\exp\left(\eta \widehat{A}_{b^n}^t(s,a)\right), & s \in \mathcal{K}^n, \\ \pi^t(\cdot|s), & else, \end{cases}$$

*where $\widehat{A}_{b^n}^t(s,a)$ is an arbitrary advantage estimator, and with $\pi^0$ initialized as:*

$$\pi^0(\cdot|s) = \begin{cases} Uniform(\mathcal{A}) & s \in \mathcal{K}^n \\ Uniform(\{a \in \mathcal{A} : (s,a) \notin \mathcal{K}^n\}) & else. \end{cases}$$

*Assume that $\sup_{s,a}\left|\widehat{A}_{b^n}^t(s,a)\right| \leq W$ and $\mathbb{E}_{a'\sim\pi_s^t}\widehat{A}_{b^n}^t(s,a') = 0$ for all $t$. Then the NPG outputs a sequence of policies $\{\pi^t\}_{t=1}^T$ such that on $\mathcal{M}^n$, when comparing to $\widetilde{\pi}^n$:*

$$\frac{1}{T}\sum_{t=1}^T \left(V_{\mathcal{M}^n}^{\widetilde{\pi}^n} - V_{\mathcal{M}^n}^t\right) = \frac{1}{T}\sum_{t=1}^T \left(V_{\mathcal{M}^n}^{\widetilde{\pi}^n} - V_{b^n}^t\right)$$

$$\leq \frac{1}{1-\gamma}\left(2W\sqrt{\frac{\log(A)}{T}} + \frac{1}{T}\sum_{t=1}^T \left(\mathbb{E}_{(s,a)\sim\widetilde{d}_{\mathcal{M}^n}} \left(A_{b^n}^t(s,a) - \widehat{A}_{b^n}^t(s,a)\right) \mathbf{1}\{s \in \mathcal{K}^n\}\right)\right),$$

*Proof.* First consider any policy $\pi$ which uniformly picks actions among $\{a \in \mathcal{A} : (s,a) \notin \mathcal{K}^n\}$ at any $s \notin \mathcal{K}^n$. Via performance difference lemma, we have:

$$V_{\mathcal{M}^n}^{\widetilde{\pi}^n} - V_{\mathcal{M}^n}^{\pi} = \frac{1}{1-\gamma}\sum_{(s,a)}\widetilde{d}_{\mathcal{M}^n}(s,a)A_{\mathcal{M}^n}^{\pi}(s,a) \leq \frac{1}{1-\gamma}\sum_{(s,a)}\widetilde{d}_{\mathcal{M}^n}(s,a)A_{\mathcal{M}^n}^{\pi}(s,a)\mathbf{1}\{s \in \mathcal{K}^n\},$$

where the last inequality comes from the fact that $A_{\mathcal{M}^n}^{\pi}(s,a)\mathbf{1}\{s \notin \mathcal{K}^n\} \leq 0$. To see this, first note that that for any $s \notin \mathcal{K}^n$, $\widetilde{\pi}^n$ will deterministically pick $a^{\dagger}$, and $Q_{\mathcal{M}^n}^{\pi}(s,a^{\dagger}) = 1 + \gamma V_{\mathcal{M}^n}^{\pi}(s)$ as taking $a^{\dagger}$ leads the agent back to $s$. Second, since $\pi$ uniformly picks actions among $\{a : (s,a) \notin \mathcal{K}^n\}$, we have $V_{\mathcal{M}_n}^{\pi} \geq 1/(1-\gamma)$ as the reward bonus $b^n(s,a)$ on $(s,a) \notin \mathcal{K}^n$ is $1/(1-\gamma)$. Hence, we have

$$A_{\mathcal{M}^n}^{\pi}(s,a^{\dagger}) = Q_{\mathcal{M}^n}^{\pi}(s,a^{\dagger}) - V_{\mathcal{M}^n}^{\pi}(s) = 1 - (1-\gamma)V_{\mathcal{M}^n}^{\pi}(s) \leq 0, \quad \forall s \notin \mathcal{K}^n.$$

Recall Algorithm 3, $\pi^t$ chooses actions uniformly randomly among $\{a : (s,a) \notin \mathcal{K}^n\}$ for $s \notin \mathcal{K}^n$, thus we have:

$$(1-\gamma)\left(V_{\mathcal{M}^n}^{\widetilde{\pi}^n} - V_{\mathcal{M}^n}^t\right) \leq \sum_{(s,a)}\widetilde{d}_{\mathcal{M}^n}(s,a)A_{\mathcal{M}^n}^t(s,a)\mathbf{1}\{s \in \mathcal{K}^n\} = \sum_{(s,a)}\widetilde{d}_{\mathcal{M}^n}(s,a)A_{b^n}^t(s,a)\mathbf{1}\{s \in \mathcal{K}^n\},$$

where the last equation uses the fact that $A_{b^n}^t(s,a) = A_{\mathcal{M}^n}^t(s,a)$ for $a \neq a^{\dagger}$ and the fact that for $s \in \mathcal{K}^n$, $\widetilde{\pi}^n$ never picks $a^{\dagger}$ (i.e., $\widetilde{d}_{\mathcal{M}^n}(s,a^{\dagger}) = 0$ for $s \in \mathcal{K}^n$).

Recall the update rule of NPG,

$$\pi^{t+1}(\cdot|s) \propto \pi^t(\cdot|s)\exp\left(\eta\left(\widehat{A}_{b^n}^t(s,\cdot)\right)\mathbf{1}\{s \in \mathcal{K}^n\}\right), \forall s,$$

which is equivalent to updating $s \in \mathcal{K}^n$ while holding $\pi(\cdot|s)$ fixed for $s \notin \mathcal{K}^n$, i.e.,

$$\pi^{t+1}(\cdot|s) \propto \begin{cases} \pi^t(\cdot|s)\exp\left(\eta\widehat{A}_{b^n}^t(s,\cdot)\right), & s \in \mathcal{K}^n, \\ \pi^t(\cdot|s), & else. \end{cases}$$

Now let us focus on any $s \in \mathcal{K}^n$. Denote the normalizer $z^t = \sum_a \pi^t(a|s) \exp\left(\eta \widehat{A}_{b^n}^t(s,a)\right)$. Note that we sum over all the actions as the state is known, so that for all the actions $a$, we have $(s,a) \in \mathcal{K}^n$. We have that:

$$\text{KL}(\widetilde{\pi}_s^n, \pi_s^{t+1}) - \text{KL}(\widetilde{\pi}_s^n, \pi_s^t) = \mathbb{E}_{a \sim \widetilde{\pi}_s^n}\left[-\eta \widehat{A}_{b^n}^t(s,a) + \log(z^t)\right],$$

where we use $\pi_s$ as a shorthand for the vector of probabilities $\pi(\cdot|s)$ over actions, given the state $s$. For $\log(z^t)$, using the assumption that $\eta \leq 1/W$, we have that $\eta \widehat{A}_{b^n}^t(s,a) \leq 1$, which allows us to use the inequality $\exp(x) \leq 1 + x + x^2$ for any $x \leq 1$ and leads to the following inequality:

$$\begin{aligned}
\log(z^t) &= \log\left(\sum_a \pi^t(a|s) \exp(\eta \widehat{A}_{b^n}^t(s,a))\right) \\
&\leq \log\left(\sum_a \pi^t(a|s)\left(1 + \eta \widehat{A}_{b^n}^t(s,a) + \eta^2\left(\widehat{A}_{b^n}^t(s,a)\right)^2\right)\right) \\
&= \log\left(1 + \eta^2 W^2\right) \leq \eta^2 W^2,
\end{aligned}$$

where we use the fact that $\sum_a \pi^t(a|s)\widehat{A}_{b^n}^t(s,a) = 0$.

Hence, for $s \in \mathcal{K}^n$ we have:

$$\text{KL}(\widetilde{\pi}_s^n, \pi_s^{t+1}) - \text{KL}(\widetilde{\pi}_s^n, \pi_s^t) \leq -\eta \mathbb{E}_{a \sim \widetilde{\pi}_s^n}\widehat{A}_{b^n}^t + \eta^2 W^2.$$

Adding terms across rounds, and using the telescoping sum, we get:

$$\sum_{t=1}^T \mathbb{E}_{a \sim \widetilde{\pi}_s^n}\widehat{A}_{b^n}^t(s,a) \leq \frac{1}{\eta}\text{KL}(\widetilde{\pi}_s^n, \pi_s^1) + \eta T W^2 \leq \frac{\log(A)}{\eta} + \eta T W^2, \quad \forall s \in \mathcal{K}^n.$$

Add $\mathbb{E}_{s \sim \widetilde{d}_{\mathcal{M}^n}}$, we have:

$$\sum_{t=1}^T \mathbb{E}_{(s,a) \sim \widetilde{d}_{\mathcal{M}^n}}\left[\widehat{A}_{b^n}^t(s,a)\mathbf{1}\{s \in \mathcal{K}^n\}\right] \leq \frac{\log(A)}{\eta} + \eta T W^2 \leq 2W\sqrt{\log(A)T}.$$

Hence, for regret on $\mathcal{M}_n$, we have:

$$\sum_{t=1}^T \left(V_{\mathcal{M}^n}^{\widetilde{\pi}^n} - V_{\mathcal{M}^n}^t\right)$$

$$\leq \sum_{t=1}^T \mathbb{E}_{(s,a) \sim \widetilde{d}_{\mathcal{M}^n}}\left[\widehat{A}_{b^n}^t(s,a)\mathbf{1}\{s \in \mathcal{K}^n\}\right] + \sum_{t=1}^T \left(\mathbb{E}_{(s,a) \sim \widetilde{d}_{\mathcal{M}^n}}\left(A_{b^n}^t(s,a) - \widehat{A}_{b^n}^t(s,a)\right)\mathbf{1}\{s \in \mathcal{K}^n\}\right)$$

$$\leq 2W\sqrt{\log(A)T} + \sum_{t=1}^T \left(\mathbb{E}_{(s,a) \sim \widetilde{d}_{\mathcal{M}^n}}\left(A_{b^n}^t(s,a) - \widehat{A}_{b^n}^t(s,a)\right)\mathbf{1}\{s \in \mathcal{K}^n\}\right).$$

Now using the fact that $\pi^t$ never picks $a^\dagger$, we have $V_{\mathcal{M}^n}^t = V_{b^n}^t$. This concludes the proof. $\qquad\square$

Note that the second term of the RHS of the inequality in the above lemma measures the average estimation error of $\widehat{A}_{b^n}^t$. Below, for PC-PG's analysis, we bound the critic prediction error under $\widetilde{d}_{\mathcal{M}^n}$.

## C  Relationship between $\mathcal{M}^n$ and $\mathcal{M}$

We need the following lemma to relate the probability of a known state being visited by $\widetilde{\pi}^n$ under $\mathcal{M}^n$ and the probability of the same state being visited by $\widetilde{\pi}$ under $\mathcal{M}_{b^n}$. Note that intuitively as $\widetilde{\pi}^n$ always picks $a^\dagger$ outside $\mathcal{K}^n$, it should have smaller probability of visiting the states inside $\mathcal{K}^n$ (once $\widetilde{\pi}^n$ escapes, it will be absorbed and will never return back to $\mathcal{K}^n$). Also recall that $\mathcal{M}_{b^n}$ and $\mathcal{M}$ share the same underlying transition dynamics. So for any policy, we simply have $d_{\mathcal{M}_{b^n}}^\pi = d^\pi$.

The following lemma formally states this.

**Lemma C.1.** *Consider any state $s \in \mathcal{K}^n$, we have:*

$$\widetilde{d}_{\mathcal{M}^n}(s,a) \le d^{\widetilde{\pi}}(s,a), \forall a \in \mathcal{A},$$

*where recall $\widetilde{d}_{\mathcal{M}^n}$ is the state-action distribution of $\widetilde{\pi}^n$ under $\mathcal{M}^n$.*

*Proof.* We prove by induction. Recall $\widetilde{d}_{\mathcal{M}^n}$ is the state-action distribution of $\widetilde{\pi}^n$ under $\mathcal{M}^n$, and $d^{\widetilde{\pi}}$ is the state-action distribution of $\widetilde{\pi}$ under both $\mathcal{M}_{b^n}$ and $\mathcal{M}$ as they share the same dynamics. For any $h$, let $\widetilde{d}_{\mathcal{M}^n,h}$ further denote the state-action distribution of $\widetilde{\pi}^n$ under $\mathcal{M}^n$ after taking $h$ steps, starting from $s_0$, and let $d_h^{\widetilde{\pi}}$ refer to the same quantity in the original MDP $\mathcal{M}$.

Starting at $h = 0$, we have:

$$\widetilde{d}_{\mathcal{M}^n,0}(s_0, a) = d_0^{\widetilde{\pi}}(s_0, a),$$

as $s_0$ is fixed and $s_0 \in \mathcal{K}^n$, and $\widetilde{\pi}^n(\cdot|s_0) = \widetilde{\pi}(\cdot|s_0)$.

Now assume that at time step $h$, we have that for all $s \in \mathcal{K}^n$, we have:

$$\widetilde{d}_{\mathcal{M}^n,h}(s,a) \le d_h^{\widetilde{\pi}}(s,a), \forall a \in \mathcal{A}.$$

Now we proceed to prove that this holds for $h + 1$. By definition, we have that for $s \in \mathcal{K}^n$,

$$\widetilde{d}_{\mathcal{M}^n,h+1}(s) = \sum_{s',a'} \widetilde{d}_{\mathcal{M}^n,h}(s',a') P_{\mathcal{M}^n}(s|s',a')$$

$$= \sum_{s',a'} \mathbf{1}\{s' \in \mathcal{K}^n\} \widetilde{d}_{\mathcal{M}^n,h}(s',a') P_{\mathcal{M}^n}(s|s',a') = \sum_{s',a'} \mathbf{1}\{s' \in \mathcal{K}^n\} \widetilde{d}_{\mathcal{M}^n,h}(s',a') P(s|s',a')$$

as if $s' \notin \mathcal{K}^n$, $\widetilde{\pi}^n$ will deterministically pick $a^\dagger$ (i.e., $a' = a^\dagger$) and $P_{\mathcal{M}^n}(s|s',a^\dagger) = 0$.

On the other hand, for $d_{h+1}^{\widetilde{\pi}}(s,a)$, we have that for $s \in \mathcal{K}^n$,

$$d_{h+1}^{\widetilde{\pi}}(s,a) = \sum_{s',a'} d_h^{\widetilde{\pi}}(s',a') P(s|s',a')$$

$$= \sum_{s',a'} \mathbf{1}\{s' \in \mathcal{K}^n\} d_h^{\widetilde{\pi}}(s',a') P(s|s',a') + \sum_{s',a'} \mathbf{1}\{s \notin \mathcal{K}^n\} d_h^{\widetilde{\pi}}(s',a') P(s|s',a')$$

$$\ge \sum_{s',a'} \mathbf{1}\{s' \in \mathcal{K}^n\} d_h^{\widetilde{\pi}}(s',a') P(s|s',a')$$

$$\ge \sum_{s',a'} \mathbf{1}\{s' \in \mathcal{K}^n\} \widetilde{d}_{\mathcal{M}^n,h}(s',a') P(s|s',a') = \widetilde{d}_{\mathcal{M}^n,h+1}(s).$$

Using the fact that $\widetilde{\pi}^n(\cdot|s) = \widetilde{\pi}(\cdot|s)$ for $s \in \mathcal{K}^n$, we conclude that the inductive hypothesis holds at $h + 1$ as well. Thus it holds for all $h$. Using the definition of average state-action distribution, we conclude the proof. $\square$

We now establish a standard simulation lemma-style result to link the performance of policies on $\mathcal{M}^n$ to the performance on the real MDP $\mathcal{M}$, before bounding the error in the lemma using a linear bandits potential function argument as sketched above. These arguments allow us to translate the error bounds from Appendix B from the augmented MDP $\mathcal{M}^n$ to the actual MDP $\mathcal{M}$.

**Lemma C.2** (Policy performance on $\mathcal{M}^n$, $\mathcal{M}_{b^n}$ $\mathcal{M}$)**.** *At each episode $n$, denote $\{\pi^t\}_{t=1}^T$ as the sequence of policies generated from NPG (Algorithm 3) in that episode. We have that for $\widetilde{\pi}^n$ and $\pi^t$ for any $t \in [T]$:*

$$V_{\mathcal{M}^n}^{\widetilde{\pi}^n} \ge V_{\mathcal{M}}^{\widetilde{\pi}},$$

$$V_{\mathcal{M}}^t \ge V_{b^n}^t - \frac{1}{1-\gamma} \left( \sum_{(s,a) \notin \mathcal{K}^n} d^t(s,a) \right).$$

*Proof.* Note that when running $\widetilde{\pi}^n$ under $\mathcal{M}^n$, once $\widetilde{\pi}^n$ visits $s \notin \mathcal{K}^n$, it will be absorbed into $s$ and keeps looping there and receiving the maximum reward 1. Note that $\widetilde{\pi}$ receives reward no more than 1 and in $\mathcal{M}$ we do not have reward bonus.

Recall that $\pi^t$ never takes $a^\dagger$. Hence $d^t(s,a) = d^t_{\mathcal{M}_{b^n}}(s,a)$ for all $(s,a)$. Recall that the reward bonus is defined as $\frac{1}{1-\gamma}\mathbf{1}\{(s,a) \notin \mathcal{K}^n\}$. Using the definition of $b^n(s,a)$ concludes the proof. $\square$

The lemma below relates the escaping probability to an elliptical potential function and quantifies the progress made by the algorithm by the maximum information gain quantity.

**Lemma C.3** (Potential Function Argument). *Consider the sequence of policies* $\{\pi^n\}_{n=1}^N$ *generated from* Algorithm 2. *We have:*

$$\sum_{n=0}^{N-1} V^{\pi^{n+1}} \geq \sum_{n=0}^{N-1} V_{b^n}^{\pi^{n+1}} - \frac{2\mathcal{I}_N(\lambda)}{\beta(1-\gamma)}$$

*Proof.* Denote the eigen-decomposition of $\Sigma_{\mathrm{cov}}^n$ as $U\Lambda U^\top$ and $\Sigma^n = \mathbb{E}_{(s,a)\sim d^n}\phi\phi^\top$. We have:

$$\mathrm{tr}\left(\Sigma^{n+1}\left(\Sigma_{\mathrm{cov}}^n\right)^{-1}\right) = \mathbb{E}_{(s,a)\sim d^{n+1}}\,\mathrm{tr}\left(\phi(s,a)\phi(s,a)^\top\left(\Sigma_{\mathrm{cov}}^n\right)^{-1}\right)$$

$$= \mathbb{E}_{(s,a)\sim d^{n+1}}\phi(s,a)^\top\left(\Sigma_{\mathrm{cov}}^n\right)^{-1}\phi(s,a)$$

$$\geq \mathbb{E}_{(s,a)\sim d^{n+1}}\left[\mathbf{1}\{(s,a) \notin \mathcal{K}^n\}\phi(s,a)^\top\left(\Sigma_{\mathrm{cov}}^n\right)^{-1}\phi(s,a)\right] \geq \beta\mathbb{E}_{(s,a)\sim d^{n+1}}\mathbf{1}\{(s,a)\notin\mathcal{K}^n\}$$

together with Lemma C.2, which implies that

$$V_{b^n}^{\pi^{n+1}} - V^{\pi^{n+1}} \leq \frac{\mathrm{tr}\left(\Sigma^{n+1}\left(\Sigma_{\mathrm{cov}}^n\right)^{-1}\right)}{\beta(1-\gamma)}.$$

Now using Lemma H.2, we have:

$$\sum_{n=0}^{N}\left(V_{b^n}^{\pi^{n+1}} - V^{\pi^{n+1}}\right) \leq \frac{2\log(\det\left(\Sigma_{\mathrm{cov}}^N\right)/\det(\lambda I))}{\beta(1-\gamma)} \leq \frac{2\mathcal{I}_N(\lambda)}{\beta(1-\gamma)}$$

where we use the definition of information gain $\mathcal{I}_N(\lambda)$. $\square$

# D    Analysis of PC-PG for the Agnostic Setting (Theorem 4.3)

In this section, we analyze the performance of PC-PG using the NPG results we derived from the previous section. We begin with an assumption and a theorem statement which is the most general sample complexity result for PC-PG and from which all the statements of Section 4 follow.

We recall Assumption 4.1 which is assumed as the only structural assumption throughout this section.

The following theorem states the detailed sample complexity of PC-PG (a detailed version of Theorem 4.3).

**Theorem D.1** (Main Result: Sample Complexity of PC-PG). *Fix $\delta \in (0, 1/2)$ and $\epsilon \in (0, \frac{1}{1-\gamma})$. Setting hyperparameters as follows:*

$$T = \frac{4W^2\log(A)}{(1-\gamma)^2\epsilon^2}, \quad \lambda = 1, \quad \beta = \frac{\epsilon^2(1-\gamma)^2}{4W^2}, \quad N \geq \frac{4W^2\mathcal{I}_N(1)}{(1-\gamma)^3\epsilon^3},$$

$$M = \frac{144W^4\mathcal{I}_N(1)^2\ln(NT/\delta)}{\epsilon^6(1-\gamma)^{10}}, \quad K = 32N^2\log\left(\frac{N\widehat{d}}{\delta}\right),$$

*Under* Assumption 4.1, *with probability at least $1 - 2\delta$, we have:*

$$\max_{n\in[N]} V^{\pi^n} \geq V^{\widetilde{\pi}} - \frac{2\sqrt{A\varepsilon_{bias}}}{1-\gamma} - 4\epsilon,$$

*for any comparator $\widetilde{\pi} \in \Pi_{linear}$, with at most total number of samples:*

$$\frac{c\nu W^8 \mathcal{I}_N(1)^3 \ln(A)}{\epsilon^{11}(1-\gamma)^{15}},$$

*where $c$ is a universal constant, and $\nu$ contains only log terms:*

$$\nu = \ln\left(\frac{4AW^2 \mathcal{I}_N^2(1)}{(1-\gamma)^4 \epsilon^3 \delta c_0^2}\right) + \ln\left(\frac{16W^4 \ln(A)\mathcal{I}_N(1)}{\epsilon^5(1-\gamma)^5 \delta}\right).$$

**Remark D.1.** Note that in the above theorem, we require that the number of iterations $N$ to satisfy the constraint $N \geq 4W^2 \mathcal{I}_N(1)/((1-\gamma)^3 \epsilon^3)$. The specific $N$ thus depends on the form of the maximum information gain $\mathcal{I}_N(1)$. For instance, when $\phi(s,a) \in \mathbb{R}^d$ with $\|\phi\|_2 \leq 1$, we have $\mathcal{I}_N(1) \leq d \log(N+1)$. Hence setting $N \geq \frac{8W^2 d}{(1-\gamma)^3 \epsilon^3} \ln\left(\frac{4W^2 d}{(1-\gamma)^3 \epsilon^3}\right)$ suffices. Another example is when $\phi$ lives in an RKHS with RBF kernel. In this case, we have $\mathcal{I}_N(1) = O(\log(N)^{d_{s,a}})$ ([59]), where $d_{s,a}$ stands for the dimension of the concatenated vector of state and action. In this case, we can set $N = O\left(\frac{W^2}{(1-\gamma)^3 \epsilon^3}\left(\ln\left(\frac{W^2}{(1-\gamma)^3 \epsilon^3}\right)\right)^{d_{s,a}}\right)$.

In the rest of this section, we prove the theorem. Given the analysis of Appendix B, proving the theorem requires the following steps at a high-level:

1. Bounding the number of outer iterations $N$ in order to obtain a desired accuracy $\epsilon$. Intuitively, this requires showing that the probability with which we can reach an *unknown state* with a positive reward bonus is appropriately small. We carry out this bounding by using arguments from the analysis of linear bandits [19]. At a high-level, if there is a good probability of reaching unknown states, then NPG finds them based on our previous analysis as these states carry a high reward. But every time we find such states, the covariance matrix of the resulting policy contains directions not visited by the previous cover with a large probability (or else the quadratic form defining the unknown states would be small). In a $d$-dimensional linear space, the number of times we can keep finding significantly new directions is roughly $O(d)$ (or more precisely based on the intrinsic dimension), which allows us to bound the number of required outer episodes.

2. Bounding the prediction error of the critic in Lemma B.1. This can be done by a standard regression analysis and we use a specific result for stochastic gradient descent to fit the critic.

3. Errors from empirical covariance matrices instead of their population counterparts have to be accounted for as well, and this is done by using standard inequalities on matrix concentration [65].

## D.1 Proof of Theorem D.1

We recall that we perform linear regression from $\phi(s,a)$ to $Q_{b^n}^\pi(s,a) - b^n(s,a)$, and set $\widehat{A}_{b^n}^t(s,a)$ as

$$\widehat{A}_{b^n}^t(s,a) = \left(b^n(s,a) + \theta^t \cdot \phi(s,a)\right) - \mathbb{E}_{a' \sim \pi_s^t}[b^n(s,a') + \theta^t \cdot \phi(s,a')]$$
$$:= \bar{b}^{n,t}(s,a) + \theta^t \cdot \bar{\phi}^t(s,a),$$

where for notation simplicity, we denote centered bonus $\bar{b}^{n,t}(s,a) = b^n(s,a) - \mathbb{E}_{a' \sim \pi_s^t} b^n(s,a')$, and centered feature $\bar{\phi}^t(s,a) = \phi(s,a) - \mathbb{E}_{a' \sim \pi_s^t} \phi(s,a')$.

**Lemma D.1** (Variance and Bias Tradeoff)**.** *Assume that at episode $n$ we have $\phi(s,a)^\top (\Sigma_{cov}^n)^{-1} \phi(s,a) \leq \beta$ for $(s,a) \in \mathcal{K}^n$. At iteration $t$ inside episode $n$, let us denote a best on-policy fit as*

$$\theta_\star^t \in \operatorname*{argmin}_{\|\theta\| \leq W} \mathbb{E}_{(s,a) \sim \rho_{cov}^n}\left(\left(Q_{b^n}^t(s,a) - b^n(s,a)\right) - \theta \cdot \phi(s,a)\right)^2.$$

*Assume the following condition is true for all $t \in [T]$:*

$$L\left(\theta^t; \rho_{cov}^n, Q_{b^n}^t - b^n\right) \leq \min_{\theta:\|\theta\| \leq W} L\left(\theta; \rho_{cov}^n, Q_{b^n}^t - b^n\right) + \varepsilon_{stat},$$

where $\varepsilon_{stat} \in \mathbb{R}^+$. *Then under* Assumption 4.1 *(with $\widetilde{\pi}$ as the comparator policy here), we have that for all $t \in [T]$:*

$$\mathbb{E}_{(s,a)\sim \widetilde{d}_{\mathcal{M}^n}} \left( A_{b^n}^t(s,a) - \widehat{A}_{b^n}^t(s,a) \right) \mathbf{1}\{s \in \mathcal{K}^n\} \leq 2\sqrt{A\varepsilon_{bias}} + 2\sqrt{\beta\lambda W^2} + 2\sqrt{\beta n \varepsilon_{stat}}.$$

*Proof.* We first show that under the condition involving $\varepsilon_{stat}$ above, $\mathbb{E}_{(s,a)\sim \rho_{\text{cov}}^n} \left( \theta_\star^t \cdot \phi(s,a) - \theta^t \cdot \phi(s,a) \right)^2$ is bounded by $\varepsilon_{stat}$.

$$\mathbb{E}_{(s,a)\sim \rho_{\text{cov}}^n} \left( Q_{b^n}^t(s,a) - b^n(s,a) - \theta^t \cdot \phi(s,a) \right)^2 - \mathbb{E}_{(s,a)\sim \rho_{\text{cov}}^n} \left( Q_{b^n}^t(s,a) - b^n(s,a) - \theta_\star^t \cdot \phi(s,a) \right)^2$$
$$= \mathbb{E}_{(s,a)\sim \rho_{\text{cov}}^n} \left( \theta_\star^t \cdot \phi(s,a) - \theta^t \cdot \phi(s,a) \right)^2 + 2\mathbb{E}_{(s,a)\sim \rho_{\text{cov}}^n} \left( Q_{b^n}^t(s,a) - b^n(s,a) - \theta_\star^t \cdot \phi(s,a) \right) \phi(s,a)^\top \left( \theta_\star^t - \theta^t \right).$$

Note that $\theta_\star$ is one of the minimizers of the constrained square loss $\mathbb{E}_{(s,a)\sim \rho_{\text{cov}}^n}(Q_{b^n}^t(s,a) - b^n(s,a) - \theta \cdot \phi(s,a))^2$, via first-order optimality, we have:

$$\mathbb{E}_{(s,a)\sim \rho_{\text{cov}}^n} \left( Q_{b^n}^t(s,a) - b^n(s,a) - \theta_\star^t \cdot \phi(s,a) \right) \left( -\phi(s,a)^\top \right) \left( \theta - \theta_\star^t \right) \geq 0,$$

for any $\|\theta\| \leq W$, which implies that:

$$\mathbb{E}_{(s,a)\sim \rho_{\text{cov}}^n} \left( \theta_\star^t \cdot \phi(s,a) - \theta^t \cdot \phi(s,a) \right)^2$$
$$\leq \mathbb{E}_{(s,a)\sim \rho_{\text{cov}}^n} \left( Q_{b^n}^t(s,a) - b^n(s,a) - \theta^t \cdot \phi(s,a) \right)^2 - \mathbb{E}_{(s,a)\sim \rho_{\text{cov}}^n} \left( Q_{b^n}^t(s,a) - b^n(s,a) - \theta_\star^t \cdot \phi(s,a) \right)^2 \leq \varepsilon_{stat}.$$

Recall that $\Sigma_{\text{cov}}^n = \sum_{i=1}^n \mathbb{E}_{(s,a)\sim d^n} \phi(s,a)\phi(s,a)^\top + \lambda \mathbf{I} = n \left( \mathbb{E}_{(s,a)\sim \rho_{\text{cov}}^n} \phi(s,a)\phi(s,a)^\top + \lambda/n \mathbf{I} \right)$.
Denote $\bar{\Sigma}_{\text{cov}}^n = \Sigma_{\text{cov}}^n/n$. We have:

$$\left( \theta_\star^t - \theta^t \right)^\top \left( \mathbb{E}_{(s,a)\sim \rho_{\text{cov}}^n} \phi(s,a)\phi(s,a)^\top + \lambda/n \mathbf{I} \right) \left( \theta_\star^t - \theta^t \right) \leq \varepsilon_{stat} + \frac{\lambda}{n} W^2.$$

Hence for any $(s,a) \in \mathcal{K}^n$, we must have the following point-wise estimation error:

$$\left| \phi(s,a)^\top \left( \theta_\star^t - \theta^t \right) \right| \leq \|\phi(s,a)\|_{(\Sigma_{\text{cov}}^n)^{-1}} \|\theta_\star^t - \theta^t\|_{\Sigma_{\text{cov}}^n} \leq \sqrt{\beta n \varepsilon_{stat} + \beta\lambda W^2}. \tag{7}$$

Now we bound $\mathbb{E}_{(s,a)\sim \widetilde{d}_{\mathcal{M}^n}} \left( A_{b^n}^t(s,a) - \widehat{A}_{b^n}^t(s,a) \right) \mathbf{1}\{s \in \mathcal{K}^n\}$ as follows.

$$\mathbb{E}_{(s,a)\sim \widetilde{d}_{\mathcal{M}^n}} \left( A_{b^n}^t(s,a) - \widehat{A}_{b^n}^t(s,a) \right) \mathbf{1}\{s \in \mathcal{K}^n\}$$
$$= \underbrace{\mathbb{E}_{(s,a)\sim \widetilde{d}_{\mathcal{M}^n}} \left( A_{b^n}^t(s,a) - (\bar{b}^{n,t}(s,a) + \theta_\star^t \cdot \bar{\phi}^t(s,a)) \right) \mathbf{1}\{s \in \mathcal{K}^n\}}_{\text{term A}}$$
$$+ \underbrace{\mathbb{E}_{(s,a)\sim \widetilde{d}_{\mathcal{M}^n}} \left( (\bar{b}^{n,t}(s,a) + \theta_\star^t \cdot \bar{\phi}^t(s,a)) - (\bar{b}^{n,t}(s,a) + \theta^t \cdot \bar{\phi}^t(s,a)) \right) \mathbf{1}\{s \in \mathcal{K}^n\}}_{\text{term B}}.$$

We first bound term A above.

$$\mathbb{E}_{(s,a)\sim \widetilde{d}_{\mathcal{M}^n}} \left( A_{b^n}^t(s,a) - \bar{b}^{n,t}(s,a) - \theta_\star^t \cdot \bar{\phi}^t(s,a) \right) \mathbf{1}\{s \in \mathcal{K}^n\}$$
$$= \mathbb{E}_{(s,a)\sim \widetilde{d}_{\mathcal{M}^n}} \left( Q_{b^n}^t(s,a) - b^n(s,a) - \theta_\star^t \cdot \phi(s,a) \right) \mathbf{1}\{s \in \mathcal{K}^n\}$$
$$\quad + \mathbb{E}_{s\sim \widetilde{d}_{\mathcal{M}^n}, a\sim \pi_s^t} \left( -Q_{b^n}^t(s,a) + b^n(s,a) + \theta_\star^t \cdot \phi(s,a) \right) \mathbf{1}\{s \in \mathcal{K}^n\}$$
$$\leq \sqrt{\mathbb{E}_{(s,a)\sim \widetilde{d}_{\mathcal{M}^n}} \left( Q_{b^n}^t(s,a) - b^n(s,a) - \theta_\star^t \cdot \phi(s,a) \right)^2 \mathbf{1}\{s \in \mathcal{K}^n\}}$$
$$\quad + \sqrt{\mathbb{E}_{s\sim \widetilde{d}_{\mathcal{M}^n}, a\sim \pi_s^t} \left( Q_{b^n}^t(s,a) - b^n(s,a) - \theta_\star^t \cdot \phi(s,a) \right)^2 \mathbf{1}\{s \in \mathcal{K}^n\}}$$
$$\leq \sqrt{\mathbb{E}_{(s,a)\sim d^{\widetilde{\pi}}} \left( Q_{b^n}^t(s,a) - b^n(s,a) - \theta_\star^t \cdot \phi(s,a) \right)^2 \mathbf{1}\{s \in \mathcal{K}^n\}}$$
$$\quad + \sqrt{\mathbb{E}_{s\sim d^{\widetilde{\pi}}, a\sim \pi_s^t} \left( Q_{b^n}^t(s,a) - b^n(s,a) - \theta_\star^t \cdot \phi(s,a) \right)^2 \mathbf{1}\{s \in \mathcal{K}^n\}}$$
$$\leq \sqrt{\mathbb{E}_{(s,a)\sim d^{\widetilde{\pi}}} \left( Q_{b^n}^t(s,a) - b^n(s,a) - \theta_\star^t \cdot \phi(s,a) \right)^2} + \sqrt{\mathbb{E}_{s\sim d^{\widetilde{\pi}}, a\sim \pi_s^t} \left( Q_{b^n}^t(s,a) - b^n(s,a) - \theta_\star^t \cdot \phi(s,a) \right)^2}$$
$$\leq 2\sqrt{A\epsilon_{bias}},$$

where the first inequality uses CS inequality, the second inequality uses Lemma C.1 for $s \in \mathcal{K}^n$, and the last inequality uses the change of variable over action distributions and Assumption 4.1.

Now we bound term B above. We have:

$$\mathbb{E}_{(s,a) \sim \widetilde{d}_{\mathcal{M}^n}} \left( \theta_\star^t \cdot \bar{\phi}^t(s,a) - \theta^t \cdot \bar{\phi}^t(s,a) \right) \mathbf{1}\{s \in \mathcal{K}^n\}$$

$$= \mathbb{E}_{(s,a) \sim \widetilde{d}_{\mathcal{M}^n}} \left( \theta_\star^t \phi(s,a) - \theta^t \cdot \phi(s,a) \right) \mathbf{1}\{s \in \mathcal{K}^n\}$$

$$\quad - \mathbb{E}_{s \sim \widetilde{d}_{\mathcal{M}^n}} \mathbb{E}_{a \sim \pi^t} \mathbf{1}\{s \in \mathcal{K}^n\} \left( \theta_\star^t \phi(s,a) - \theta^t \cdot \phi(s,a) \right) \leq 2\sqrt{\beta \lambda W^2} + 2\sqrt{\beta n \epsilon_{stat}},$$

where we use the point-wise estimation guarantee from inequality (7).

Combine term A and term B together, we conclude the proof. $\qquad \square$

Combine the the above lemma and Lemma B.1, we can see that as long as the on-policy critic achieves small statistical error (i.e., $\epsilon_{stat}$ is small), and our features $\phi(s,a)$ are sufficient to represent Q functions in a linear form (i.e., $\epsilon_{bias}$ is small), then we can guarantee inside episode $n$, NPG succeeds by finding a policy that has low regret with respect to the comparator $\widetilde{\pi}^n$:

$$\max_{t \in [T]} V_{b^n}^t \geq V_{\mathcal{M}^n}^{\widetilde{\pi}^n} - \frac{1}{1-\gamma} \left( 2W \sqrt{\frac{\log(A)}{T}} + 2\sqrt{A \varepsilon_{bias}} + 2\sqrt{\beta \lambda W^2} + 2\sqrt{\beta n \varepsilon_{stat}} \right). \quad (8)$$

The term that contains $\epsilon_{stat}$ comes from the statistical error induced from constrained linear regression. Note that in general, $\epsilon_{stat}$ decays at a rate of $O(1/\sqrt{M})$ with $M$ being the total number of data samples used for linear regression (line 6 in Algorithm 3), and $\epsilon_{stat}$ usually does not polynomially depend on dimension of $\phi(s,a)$ explicitly. See Lemma H.1 for an example where linear regression is solved via stochastic gradient descent.

Using Lemma C.3, now we can transfer the regret we computed under the sequence of models $\{\mathcal{M}_{b^n}\}$ to regret under $\mathcal{M}$. Recall that $V^\pi$ denotes $V^\pi(s_0)$ and $V^n$ is in short of $V^{\pi^n}$.

**Lemma D.2.** *Assume the condition in Lemma D.1 and Assumption 4.1 hold. For the sequence of policies $\{\pi^n\}_{n=1}^N$, we have:*

$$\max_{n \in [N]} V^n \geq V^{\widetilde{\pi}} - \frac{1}{1-\gamma} \left( 2W \sqrt{\frac{\log(A)}{T}} + 2\sqrt{A\varepsilon_{bias}} + 2\sqrt{\beta \lambda W^2} + 2\sqrt{\beta N \varepsilon_{stat}} + \frac{2\mathcal{I}_N(\lambda)}{N\beta} \right).$$

*Proof.* First combine Lemma B.1 and Lemma D.1, we have:

$$\frac{1}{N} \sum_{n=0}^{N-1} V_{b^n}^{n+1} \geq \frac{1}{N} \sum_{n=0}^{N-1} V_{\mathcal{M}^n}^{\widetilde{\pi}^n} - \frac{1}{1-\gamma} \left( 2W \sqrt{\frac{\log(A)}{T}} + 2\sqrt{A\varepsilon_{bias}} + 2\sqrt{\beta \lambda W^2} + 2\sqrt{\beta N \varepsilon_{stat}} \right).$$

Use Lemma C.2 and Lemma C.3, we have:

$$\frac{1}{N} \sum_{n=1}^{N} V^n \geq V^{\widetilde{\pi}} - \frac{1}{1-\gamma} \left( 2W \sqrt{\frac{\log(A)}{T}} + 2\sqrt{A\varepsilon_{bias}} + 2\sqrt{\beta \lambda W^2} + 2\sqrt{\beta N \varepsilon_{stat}} + \frac{\mathcal{I}_N(\lambda)}{N\beta} \right),$$

which concludes the proof. $\qquad \square$

The following theorem shows that setting hyperparameters properly, we can guarantee to learn a near optimal policy.

**Theorem D.2.** *Assume the conditions in Lemma D.1 and Assumption 4.1 hold. Fix $\epsilon \in (0, 1/(4(1-\gamma)))$. Setting hyperparameters as follows:*

$$T = \frac{4W^2 \log(A)}{(1-\gamma)^2 \epsilon^2}, \quad \lambda = 1, \quad \beta = \frac{\epsilon^2 (1-\gamma)^2}{4W^2},$$

$$N \geq \frac{4W^2 \mathcal{I}_N(1)}{(1-\gamma)^3 \epsilon^3}, \quad \epsilon_{stat} = \frac{\epsilon^3 (1-\gamma)^3}{4\mathcal{I}_N(1)},$$

*we have:*

$$\max_{n \in [N]} V^n \geq V^{\widetilde{\pi}} - \frac{2\sqrt{A\varepsilon_{bias}}}{1 - \gamma} - 4\epsilon.$$

*Proof.* The theorem can be easily verified by substituting the values of hyperparameters into Lemma D.2. $\qquad\square$

The above theorem indicates that we need to control the $\varepsilon_{stat}$ statistical error from linear regression to be small in the order of $\widetilde{O}\left(\epsilon^3(1-\gamma)^3\right)$. Recall that $M$ is the total number of samples we used for each linear regression. If $\varepsilon_{stat} = \widetilde{O}\left(1/\sqrt{M}\right)$, then we roughly will need $M$ to be in the order of $\widetilde{\Omega}\left(1/(\epsilon^6(1-\gamma)^6)\right)$. Note that we do on-policy fit in each iteration $t$ inside each episode $n$, thus we will pay total number of samples in the order of $M \times (TN)$.

Another source of samples is the samples used to estimate covariance matrices $\Sigma^n$. As $\phi$ could be infinite dimensional, we need matrix concentration without explicit dependency on dimension of $\phi$. Leveraging matrix Bernstein inequality with matrix intrinsic dimension, the following lemma shows concentration results of $\widehat{\Sigma}^n$ on $\Sigma^n$, and of $\widehat{\Sigma}^n_{\mathrm{cov}}$ on $\Sigma^n_{\mathrm{cov}}$.

**Lemma D.3** (Estimating Covariance Matrices). *Set $\lambda = 1$. Define $\widehat{d}$ as the maximum possible intrinsic dimension:*

$$\widehat{d} = \max_{\pi} \mathrm{tr}\left(\Sigma^{\pi}\right)/\|\Sigma^{\pi}\|,$$

*i.e., the maximum intrinsic dimension of the covariance matrix from a mixture policy. For $K \geq 32N^2 \ln\left(\widehat{d}N/\delta\right)$ (a parameter in Algorithm 2), with probability at least $1 - \delta$, for any $n \in [N]$, we have for all $x$ with $\|x\| \leq 1$,*

$$(1/2)x^{\top}\left(\Sigma^n_{cov}\right)^{-1}x \leq x^{\top}\left(\widehat{\Sigma}^n_{cov}\right)^{-1}x \leq 2x^{\top}\left(\Sigma^n_{cov}\right)^{-1}x$$

*Proof.* The proof of the above lemma is simply Lemma H.4.

$\qquad\square$

We are now ready to prove Theorem D.1.

*Proof of Theorem D.1.* Assume the event in Lemma D.3 holds. In this case, we have for all $n \in [N]$,

$$(1/2)x^{\top}\left(\Sigma^n_{\mathrm{cov}}\right)^{-1}x \leq x^{\top}\left(\widehat{\Sigma}^n_{\mathrm{cov}}\right)^{-1}x \leq 2x^{\top}\left(\Sigma^n_{\mathrm{cov}}\right)^{-1}x,$$

for all $\|x\| \leq 1$ and the total number of samples used for estimating covariance matrices is:

$$N \times K = N \times \left(32N^2 \ln\left(\widehat{d}N/\delta\right)\right) = 32N^3 \ln\left(\widehat{d}N/\delta\right) \tag{9}$$

$$= \frac{(32 \times 64)\mathcal{I}_N(1)^3 W^6}{\epsilon^9(1-\gamma)^9} \ln\left(\frac{4\widehat{d}W^2\mathcal{I}_N(1)}{(1-\gamma)^3\epsilon^3\delta}\right) = \frac{c_1\nu_1\mathcal{I}_N(1)^3 W^6}{\epsilon^9(1-\gamma)^9}, \tag{10}$$

where $c_1$ is a constant and $\nu_1$ contains log-terms, $\nu_1 := \ln\left(\frac{4\widehat{d}W^2\mathcal{I}_N(1)}{(1-\gamma)^3\epsilon^3\delta}\right)$

Since we set known state-action pair as $\phi(s,a)^{\top}\left(\widehat{\Sigma}^n_{\mathrm{cov}}\right)^{-1}\phi(s,a) \leq \beta$, then we must have that for any $(s,a) \in \mathcal{K}^n$, we have:

$$\phi(s,a)^{\top}\left(\Sigma^n_{\mathrm{cov}}\right)^{-1}\phi(s,a) \leq 2\beta,$$

and any $(s, a) \notin \mathcal{K}^n$, we have:

$$\phi(s, a)^\top (\Sigma_{\text{cov}}^n)^{-1} \phi(s, a) \geq \frac{1}{2}\beta.$$

This allows us to call Theorem D.2. From Theorem D.2, we know that we need to set $M$ (number of samples for linear regression) large enough such that

$$\varepsilon_{stat} = \frac{\epsilon^3 (1 - \gamma)^3}{4\mathcal{I}_N(1)},$$

Using Lemma H.1 for linear regression, we know that with probability at least $1 - \delta$, for any $n, t$, $\varepsilon_{stat}$ scales in the order of:

$$\varepsilon_{stat} = \sqrt{\frac{9W^4 \log(NT/\delta)}{(1-\gamma)^4 M}},$$

where we have taken union bound over all episodes $n \in [N]$ and all iterations $t \in [T]$. Now solve for $M$, we have:

$$M = \frac{144 W^4 \mathcal{I}_N(1)^2 \ln(NT/\delta)}{\epsilon^6 (1-\gamma)^{10}}$$

Considering every episode $n \in [N]$ and every iteration $t \in [T]$, we have the total number of samples needed for NPG is:

$$NT \cdot M = \frac{4W^2 \mathcal{I}_N(1)}{\epsilon^3 (1-\gamma)^3} \times \frac{4W^2 \log(A)}{(1-\gamma)^2 \epsilon^2} \times \frac{144 W^4 \mathcal{I}_N(1)^2 \ln(NT/\delta)}{\epsilon^6 (1-\gamma)^{10}}$$

$$= \frac{c_2 W^8 \mathcal{I}_N(1)^3 \ln(A)}{\epsilon^{11}(1-\gamma)^{15}} \cdot \ln\left(\frac{16W^4 \ln(A)\mathcal{I}_N(1)}{\epsilon^5 (1-\gamma)^5 \delta}\right) = \frac{c_2 \nu_2 W^8 \mathcal{I}_N(1)^3 \ln(A)}{\epsilon^{11}(1-\gamma)^{15}},$$

where $c_2$ is a positive universal constant, and $\nu_2$ only contains log terms:

$$\nu_2 = \ln\left(\frac{16W^4 \ln(A)\mathcal{I}_N(1)}{\epsilon^5 (1-\gamma)^5 \delta}\right).$$

Combining the two sources of samples, we have that the total number of samples is bounded as:

$$\frac{c_2 \nu_2 W^8 \mathcal{I}_N(1)^3 \ln(A)}{\epsilon^{11}(1-\gamma)^{15}} + \frac{c_1 \nu_2 \mathcal{I}_N(1)^3 W^6}{\epsilon^9 (1-\gamma)^9},$$

Lastly, we relate $\widehat{d}$ to the information gain as follows. Consider a policy $\pi$ and denote the eigenvalues of $\Sigma^\pi$ as $\sigma_1 \geq \sigma_2, \dots$, where we have $\sigma_i \leq 1$.

$$\text{tr}(\Sigma^\pi)/\|\Sigma^\pi\| = \sum_i \sigma_i/\sigma_1 \leq 2\frac{1}{\sigma_1}\sum_i \ln(1 + \sigma_i) = \frac{2}{\sigma_1}\ln\det(I + \Sigma^\pi) \leq \frac{2\mathcal{I}_N(1)}{\sigma_1}.$$

We further lower bound $\sigma_1$ using our assumption on the norm $\phi(s_0, a)$ at the initial state $s_0$, i.e., $\|\phi(s_0, a)\|_2 \geq c_0$ for $c_0 \in (0, 1]$, for all $a \in \mathcal{A}$. For any $x$ with $\|x\|_2 \leq 1$, we have:

$$x^\top \Sigma^\pi x \geq (1 - \gamma)\mathbb{E}_{a \sim \pi(\cdot|s_0)} x^\top \phi(s_0, a)\phi(s_0, a)^\top x \geq \frac{1 - \gamma}{A} x^\top \phi(s_0, a')\phi(s_0, a')^\top x,$$

where in the last inequality we use the fact that there must exist an action $a'$ such that $\pi(a'|s_0) \geq 1/A$ due to $\pi(\cdot|s_0)$ is a distribution over $\mathcal{A}$. Thus, we have:

$$\sigma_1 = \max_{x: \|x\|_2 \leq 1} x^\top \Sigma^\pi x \geq \frac{1 - \gamma}{A} \max_{x: \|x\|_2 \leq 1} x^\top \phi(s_0, a')\phi(s_0, a')^\top x = \frac{(1 - \gamma)c_0^2}{A}.$$

Thus we have:

$$\widehat{d} = \max_\pi \text{tr}(\Sigma^\pi)/\|\Sigma^\pi\| \leq \frac{2A}{(1-\gamma)c_0^2}\mathcal{I}_N(1).$$

Plug in the above upper bound of $\widehat{d}$ into Eq. 9 concludes the proof. $\qquad\square$

# E    Analysis of PC-PG for Linear MDPs (Theorem 4.1)

For a linear MDP $\mathcal{M}$, recall that we assume the following parameters' norms are bounded:

$$\|v^\top \mu\| \leq \xi \in \mathbb{R}^+, \quad \|\theta\| \leq \omega \in \mathbb{R}^+, \quad \forall v, \text{ s.t. } \|v\|_\infty \leq 1.$$

With these bounds on linear MDP's parameters, we can show that for any policy $\pi$, we have $Q^\pi(s,a) = w^\pi \cdot \phi(s,a)$, with $\|w^\pi\| \leq \omega + V_{\max}\xi$, where $V_{\max} = \max_{\pi,s} V^\pi(s)$ is the maximum possible expected total value ($V_{\max}$ is at most $r_{\max}/(1-\gamma)$ with $r_{\max}$ being the maximum possible immediate reward).

At every episode $n$, recall that NPG is optimizing the MDP $\mathcal{M}_{b^n} = \{P, r(s,a) + b^n(s,a)\}$ with $P, r$ being the true transition and reward of $\mathcal{M}$ which is linear under $\phi(s,a)$.

Due to the reward bonus $b^n(s,a)$ in $\mathcal{M}_{b^n}$, $\mathcal{M}_{b^n}$ is not necessarily a linear MDP under $\phi(s,a)$ ($P$ is still linear under $\phi$ but $r(s,a) + b^n(s,a)$ it not linear anymore). Here we leverage an observation that we know $b^n(s,a)$ (as we designed it), and $Q^\pi(s,a; r + b^n) - b^n(s,a)$ is linear with respect to $\phi$ for any $(s,a) \in \mathcal{S} \times \mathcal{A}$. The following claim state this observation formally.

**Claim E.1** (Linear Property of $(Q^\pi(s,a; r + b^n) - b^n(s,a))$ under $\phi$). Consider any policy $\pi$ and any reward bonus $b^n(s,a) \in [0, 1/(1-\gamma)]$. We have that:

$$Q^\pi(s,a; r + b^n) - b^n(s,a) = w \cdot \phi(s,a), \forall s,a.$$

Further we have $\|w\| \leq \omega + \xi/(1-\gamma)^2$.

*Proof.* By definition of $Q$-function, we have:

$$Q^\pi(s,a; r + b^n) = r(s,a) + b^n(s,a) + \gamma\phi(s,a)^\top \sum_{s'} \mu(s')V^\pi(s'; r + b^n)$$

$$= b^n(s,a) + \phi(s,a) \cdot \left(\theta + \gamma\mu^\top V^\pi(\cdot; r + b^n)\right) := b^n(s,a) + \phi(s,a) \cdot w,$$

where note that $w$ is independent of $(s,a)$. Rearrange terms, we prove that $Q^\pi(s,a; r + b^n) - b^n(s,a) = w \cdot \phi(s,a)$.

Further, using the norm bounds we have for $\theta$ and $\mu$, and the fact that $\|V^\pi(\cdot; r + b^n)\|_\infty \leq 1/(1-\gamma)^2$, we conclude the proof. $\square$

The above claim supports our specific choice of critic $\widehat{A}_{b^n}^t$ in the algorithm, where we recall that we perform linear regression from $\phi(s,a)$ to $Q_{b^n}^\pi(s,a) - b^n(s,a)$, and set $\widehat{A}_{b^n}^t(s,a)$ as

$$\widehat{A}_{b^n}^t(s,a) = \left(b^n(s,a) + \theta^t \cdot \phi(s,a)\right) - \mathbb{E}_{a' \sim \pi_s^t}[b^n(s,a') + \theta^t \cdot \phi(s,a')]$$

$$:= \bar{b}^{n,t}(s,a) + \theta^t \cdot \bar{\phi}^t(s,a),$$

where $\bar{b}^{n,t}(s,a) = b^n(s,a) - \mathbb{E}_{a' \sim \pi_s^t} b^n(s,a')$, and $\bar{\phi}^t(s,a) = \phi(s,a) - \mathbb{E}_{a' \sim \pi_s^t}\phi(s,a')$.

We now prove Theorem 4.1 by showing that $\epsilon_{bias}$ is zero.

**Lemma E.1.** *Consider Assumption 4.1. For any episode $n$, iteration $t$, we have $\epsilon_{bias} = 0$.*

*Proof.* At iteration $t$, denote $\theta_\star^t$ as the linear parameterization of $Q_{b^n}^{\pi^t}(s,a) - b^n(s,a)$, i.e.,

$$\theta_\star^t \cdot \phi(s,a) = Q_{b^n}^{\pi^t}(s,a) - b^n(s,a), \quad \forall s,a,$$

where the existence of $\theta_\star^t$ follows by Claim E.1. We know that

$$\theta_\star^t \in \underset{\theta:\|\theta\| \leq W}{\operatorname{argmin}} L(\theta; \rho_{\text{cov}}^n, Q_{b^n}^t - b^n),$$

as $L(\theta_\star^t; \rho_{\text{cov}}^n, Q_{b^n}^t - b^n) = 0$. This indicates that $\theta_\star^t$ is one of the best on-policy fits. Now when transfer $\theta_t^\star$ to a different distribution $d^{\pi^\star} \circ \text{Unif}_{\mathcal{A}}$, we simply have:

$$\mathbb{E}_{(s,a) \sim d^{\pi^\star} \circ \text{Unif}_{\mathcal{A}}} \left(\theta_\star^t \cdot \phi(s,a) - \left(Q_{b^n}^t(s,a) - b^n(s,a)\right)\right)^2 = 0.$$

This concludes the proof. $\square$

We can now conclude the proof of Theorem 4.1 by invoking Theorem 4.3 with $\epsilon_{bias} = 0$. $\square$

# F   Analysis of PC-PG for State-Aggregation (Theorem 4.2)

In this section, we analyze Theorem 4.2 for state-aggregation. Similar to the analysis for linear MDP, we provide a bias-variance tradeoff lemma that is analogous to Lemma D.1. However, unlike linear MDP, here due to model-misspecification from state-aggregation, the transfer error $\epsilon_{bias}$ will not be zero. But we will show that the transfer error is related to a term that is an expected model-misspecification averaged over a fixed comparator's state distribution.

First recall the definition of state aggregation $\phi : \mathcal{S} \times \mathcal{A} \to \mathcal{Z}$. We abuse the notation a bit, and denote $\phi(s,a) = \mathbf{1}\{\phi(s,a) = z\} \in \mathbb{R}^{|\mathcal{Z}|}$, i.e., the feature vector $\phi$ indicates which $z$ the state action pair $(s,a)$ is mapped to. The following claim studies the approximation of $Q$-values under state aggregation.

**Claim F.1.** Consider any MDP with transition $P$ and reward $r$. Denote aggregation error $\epsilon_z$ as:

$$\max\left\{\|P(\cdot|s,a) - P(\cdot|s',a')\|_1, |r(s,a) - r(s',a')|\right\} \leq \epsilon_z, \forall(s,a),(s',a'), \text{ s.t.}, \phi(s,a) = \phi(s',a') = z.$$

Then, for any policy $\pi$, $(s,a)$, $(s',a')$, $z$, such that $\phi(s,a) = \phi(s',a') = z$, we have:

$$|Q^\pi(s,a) - Q^\pi(s',a')| \leq \frac{r_{\max}\epsilon_z}{1-\gamma},$$

where $r(s,a) \in [0, r_{\max}]$ for $r_{\max} \in \mathbb{R}^+$.

*Proof.* Starting from the definition of $Q^\pi$, we have:

$$|Q^\pi(s,a) - Q^\pi(s',a')| = |r(s,a) - r(s',a')| + \gamma|\mathbb{E}_{x' \sim P_{s,a}} V^\pi(x') - \mathbb{E}_{x' \sim P_{s',a'}} V^\pi(x')|$$

$$\leq \epsilon_z + \frac{r_{\max}\gamma}{1-\gamma}\|P_{s,a} - P_{s',a'}\|_1 \leq \frac{r_{\max}\epsilon_z}{1-\gamma},$$

where we use the assumption that $\phi(s,a) = \phi(s',a') = z$, and the fact that value function $\|V\|_\infty \leq r_{\max}/(1-\gamma)$ as $r(s,a) \in [0, r_{\max}]$. $\square$

Now we state the bias and variance tradeoff lemma for state aggregation.

**Lemma F.1** (Bias and Variance Tradeoff for State Aggregation). *Set $W := \sqrt{|\mathcal{Z}|}/(1-\gamma)^2$. Consider any episode $n$. Assume that we have $\phi(s,a)^\top \left(\Sigma_{cov}^n\right)^{-1}\phi(s,a) \leq \beta \in \mathbb{R}^+$ for $(s,a) \in \mathcal{K}^n$, and the following condition is true for all $t \in \{0,\ldots,T-1\}$:*

$$L^t(\theta^t; \rho_{cov}^n, Q_{b^n}^t - b^n) \leq \min_{\theta:\|\theta\| \leq W} L^t(\theta; \rho_{cov}^n, Q_{b^n}^t - b^n) + \epsilon_{stat} \in \mathbb{R}^+.$$

*We have that for all $t \in \{0,\ldots,T-1\}$ at episode $n$:*

$$\mathbb{E}_{(s,a) \sim \widetilde{d}_{\mathcal{M}^n}} \left(A_{b^n}^t(s,a) - \widehat{A}_{b^n}^t(s,a)\right) \mathbf{1}\{s \in \mathcal{K}^n\}$$

$$\leq 2\sqrt{\beta\lambda W^2} + 2\sqrt{\beta n\epsilon_{stat}} + \frac{2\mathbb{E}_{(s,a) \sim d^{\widetilde{\pi}}} \max_{a'}\left[\epsilon_{\phi(s,a')}\right]}{(1-\gamma)^2}.$$

Note that comparing to Lemma D.1, the above lemma replaces $\sqrt{A\epsilon_{bias}}$ by the average model-misspecification $\frac{\mathbb{E}_{(s,a) \sim d^{\widetilde{\pi}}} \max_{a'}\left[\epsilon_{\phi(s,a')}\right]}{(1-\gamma)^2}$.

*Proof.* We first compute one of the minimizers of $L^t(\theta; \rho_{cov}^n, Q_{b^n}^t - b^n)$. Recall the definition of $L^t(\theta; \rho_{cov}^n, Q_{b^n}^t - b^n)$, we have:

$$\mathbb{E}_{(s,a) \sim \rho_{cov}^n} \left(\theta \cdot \phi(s,a) - Q_{b^n}^{\pi^t}(s,a) + b^n(s,a)\right)^2$$

$$= \mathbb{E}_{(s,a) \sim \rho_{cov}^n} \sum_z \mathbf{1}\{\phi(s,a) = z\}\left(\theta_z - Q_{b^n}^{\pi^t}(s,a) + b^n(s,a)\right)^2,$$

which means that for an unconstrained loss minimizer $\theta_\star^t$, we have:

$$\sum_{s,a} \rho_{\text{cov}}^t(s,a)\mathbf{1}\{\phi(s,a) = z\}\left(\theta_z - Q_{b^n}^{\pi^t}(s,a) + b^n(s,a)\right) = 0,$$

which implies that $\theta_{\star,z}^t := \frac{\sum_{s,a} \rho_{\text{cov}}^n(s,a)\mathbf{1}\{\phi(s,a)=z\}(Q_{b^n}^{\pi^t}(s,a)-b^n(s,a))}{\sum_{s,a} \rho_{\text{cov}}^n(s,a)\mathbf{1}\{\phi(s,a)=z\}}$. Note that $|\theta_{\star,z}^t| \leq \frac{1}{(1-\gamma)^2}$, hence $\|\theta_\star^t\|_2 \leq \sqrt{|\mathcal{Z}|}/(1-\gamma)^2 := W$, so that this solution is also feasible within our constraints. Hence, for any $s'', a''$ such that $\phi(s'', a'') = z$, we must have:

$$\left|\theta_{\star,z}^t - (Q_{b^n}^{\pi^t}(s'', a'') - b^n(s'', a''))\right|$$

$$= \left|\frac{\sum_{s,a} \rho_{\text{cov}}^n(s,a)\mathbf{1}\{\phi(s,a) = z\}(Q_{b^n}^{\pi^t}(s,a) - b^n(s,a))}{\sum_{s,a} \rho_{\text{cov}}^n(s,a)\mathbf{1}\{\phi(s,a) = z\}} - Q_{b^n}^{\pi^t}(s'', a'') + b^n(s'', a'')\right|$$

$$= \left|\frac{\sum_{s,a} \rho_{\text{cov}}^n(s,a)\mathbf{1}\{\phi(s,a) = z\}\left(Q_{b^n}^{\pi^t}(s,a) - Q_{b^n}^{\pi^t}(s'', a'')\right)}{\sum_{s,a} \rho_{\text{cov}}^n(s,a)\mathbf{1}\{\phi(s,a) = z\}}\right| \leq \frac{\epsilon_z}{(1-\gamma)^2},$$

where we use Claim F.1, and the fact that $r(s,a) + b^n(s,a) \in [0, 1/(1-\gamma)]$, and the fact that $b^n(s,a) = b^n(s'', a'')$ if $\phi(s,a) = \phi(s'', a'')$ as the bonus is defined under feature $\phi$.

With $\theta_\star^t$ and its optimality condition for loss $L^t(\theta; \rho_{\text{cov}}^n)$, we can prove the same point-wise estimation guarantee, i.e., for any $(s,a) \in \mathcal{K}^n$, we have:

$$\left|\phi(s,a) \cdot (\theta^t - \theta_\star^t)\right| \leq \sqrt{\beta n \epsilon_{stat} + \lambda W^2}.$$

Now we bound $\mathbb{E}_{(s,a) \sim \tilde{d}_{\mathcal{M}^n}}\left(A_{b^n}^t(s,a) - \widehat{A}_{b^n}^t(s,a)\right)\mathbf{1}\{s \in \mathcal{K}^n\}$ as follows.

$$\mathbb{E}_{(s,a) \sim \tilde{d}_{\mathcal{M}_{b^n}}}\left(A_{b^n}^t(s,a) - \widehat{A}_{b^n}^t(s,a)\right)\mathbf{1}\{s \in \mathcal{K}^n\}$$

$$= \underbrace{\mathbb{E}_{(s,a) \sim \tilde{d}_{\mathcal{M}^n}}\left(A_{b^n}^t(s,a) - \bar{b}^{t,n}(s,a) - \theta_\star^t \cdot \bar{\phi}^t(s,a)\right)\mathbf{1}\{s \in \mathcal{K}^n\}}_{\text{term A}}$$

$$+ \underbrace{\mathbb{E}_{(s,a) \sim \tilde{d}_{\mathcal{M}^n}}\left(\theta_\star^t \cdot \bar{\phi}^t(s,a) - \theta^t \cdot \bar{\phi}^t(s,a)\right)\mathbf{1}\{s \in \mathcal{K}^n\}}_{\text{term B}}.$$

Again, for term B, we can use the point-wise estimation error to bound it as:

$$\text{term B} \leq 2\sqrt{\beta \lambda W^2} + 2\sqrt{\beta n \epsilon_{stat}}.$$

For term A, we have:

$$\mathbb{E}_{(s,a) \sim \tilde{d}_{\mathcal{M}^n}}\left(A_{b^n}^t(s,a) - \bar{b}^{t,n}(s,a) - \theta_\star^t \cdot \bar{\phi}^t(s,a)\right)\mathbf{1}\{s \in \mathcal{K}^n\}$$

$$\leq \mathbb{E}_{(s,a) \sim \tilde{d}_{\mathcal{M}^n}}\left|Q_{b^n}^t(s,a) - b^n(s,a) - \theta_\star^t \cdot \phi(s,a)\right|\mathbf{1}\{s \in \mathcal{K}^n\}$$

$$\quad + \mathbb{E}_{s \sim \tilde{d}_{\mathcal{M}^n}, a \sim \pi_s^t}\left|-Q_{b^n}^t(s,a) + b^n(s,a) + \theta_\star^t \cdot \phi(s,a)\right|\mathbf{1}\{s \in \mathcal{K}^n\}$$

$$\leq \mathbb{E}_{(s,a) \sim d^{\tilde{\pi}}}\left|Q_{b^n}^t(s,a) - b^n(s,a) - \theta_\star^t \cdot \phi(s,a)\right| + \mathbb{E}_{s \sim d^{\tilde{\pi}}, a \sim \pi_s^t}\left|-Q_{b^n}^t(s,a) + b^n(s,a) + \theta_\star^t \cdot \phi(s,a)\right|,$$

where last inequality uses Lemma C.1 for $s \in \mathcal{K}^n$ to switch from $\tilde{d}_{\mathcal{M}^n}$ to $d^{\tilde{\pi}}$—the state-action distribution of the comparator $\tilde{\pi}$ in the real MDP $\mathcal{M}$.

Note that for any $d \in \mathcal{S} \times \mathcal{A}$, we have:

$$\mathbb{E}_{(s,a) \sim d}\left|Q_{b^n}^t(s,a) - b^n(s,a) - \theta_\star^t \cdot \phi(s,a)\right|$$

$$\leq \sum_z \mathbb{E}_{(s,a) \sim d}\mathbf{1}\{\phi(s,a) = z\}\left|Q_{b^n}^t(s,a) - b^n(s,a) - \theta_{\star,z}^t\right| \leq \mathbb{E}_{z \sim d}\frac{\epsilon_z}{(1-\gamma)^2} = \frac{\mathbb{E}_{(s,a) \sim d}\epsilon_{\phi(s,a)}}{(1-\gamma)^2}.$$

$r(s_0, L) = 1/2$
$\phi(s_0, L) = e_1,$
$\phi(s_0, R) = e_2,$
$\phi(s_1, a) = e_3.$

$\forall s' \in$ sub-tree, $\phi(s', a) \perp \text{span}(e_1, e_2, e_3)$

Figure 4: The binary tree example. Note that here $s_0$ and $s_1$ have features only span in the first three standard basis, and the features for states inside the binary tree (dashed) contains features in the null space of the first three standard bases. Note that the features inside the binary tree could be arbitrary complicated. Unless the feature dimension scales $\exp(H)$ with $H$ being the depth of the tree, we cannot represent this problem in linear MDPs. The on-policy nature of PC-PG ensures that it succeeds in this example. Due to the complex features and large $\ell_\infty$ model-misspecification inside the binary tree, Bellman-backup based approaches (e.g., Q-learning) cannot guarantee successes.

With this, we have:

$$\text{term A} \le \mathbb{E}_{(s,a)\sim d^{\tilde\pi}} \left| Q_{b^n}^{\pi^t}(s,a) - b^n(s,a) - \theta_\star^n \cdot \phi(s,a) \right| + \mathbb{E}_{s\sim d^{\tilde\pi}, a\sim\pi_s^t} \left| -Q_{b^n}^{\pi^t}(s,a) + b^n(s,a) + \theta_\star^t \cdot \phi(s,a) \right|$$

$$\le \mathbb{E}_{s\sim d^{\tilde\pi}} \max_a \left| Q_{\widetilde{\mathcal{M}}}^{\pi^n}(s,a) - b^n(s,a) - \theta_\star^n \cdot \phi(s,a) \right| + \mathbb{E}_{s\sim d^{\tilde\pi}} \max_a \left| -Q_{b^n}^{\pi^t}(s,a) + b^n(s,a) + \theta_\star^t \cdot \phi(s,a) \right|$$

$$\le 2 \left( \mathbb{E}_{s\sim d^{\tilde\pi}} \max_a \left| Q_{b^n}^{\pi^t}(s,a) - b^n(s,a) - \theta_\star^t \cdot \phi(s,a) \right| \right) \le \frac{2\mathbb{E}_{s\sim d^{\tilde\pi}} \max_a \left[ \epsilon_{\phi(s,a)} \right]}{(1-\gamma)^2}$$

Combine term A and term B, we conclude the proof. □

The rest of the proof of Theorem 4.2 is almost identical to the proof of Theorem D.1 with $\sqrt{A\epsilon_{bias}}$ in Theorem D.1 being replaced by $\frac{2\mathbb{E}_{s\sim d^{\tilde\pi}} \max_{a'} \left[ \epsilon_{\phi(s,a')} \right]}{(1-\gamma)^2}$. □

## G Robustness to "Delusional Bias" with Partially Well-specified Models

In this section, we provide an additional example of model misspecification where we show that PC-PG succeeds while Bellman backup based algorithms do not. The basic spirit of the example is that if our modeling assumption holds for a sub-part of the MDP, then PC-PG can compete with the best policy that only visits states in this sub-part with some additional assumptions. In contrast, prior model-based and $Q$-learning based approaches heavily rely on the modeling assumptions being globally correct, and bootstrapping-based methods fail in particular due to their susceptibility to the delusional bias problem [42].

We emphasize that this constructed MDP and class of features have the following properties:

- It is not a linear MDP; we would need the dimension to be exponential in the depth $H$, i.e. $d = \Omega(2^H)$, in order to even approximate the MDP as a linear MDP.

- We have no reason to believe that value based methods (that rely on Bellman backups, e.g., Q learning) or model based algorithms will provably succeed for this example (or simple variants of it).

- Our example will have large worst case function approximation error, i.e. the $\ell_\infty$ error in approximating $Q^\star$ will be (arbitrarily) large.

- The example can be easily modified so that the concentrability coefficient (and the distribution mismatch coefficient) of the starting distribution (or a random initial policy) will be $\Omega(2^H)$.

Furthermore, we will see that PC-PG succeeds on this example, provably.

We describe the construction below (see Figure 4 for an example). There are two actions, denoted by $L$ and $R$. At initial state $s_0$, we have $P(s_1|s_0, L) = 1$; $P(s_1|s_1, a) = 1$ for any $a \in \{L, R\}$. We set the reward of taking the left action at $s_0$ to be $1/2$, i.e. $r(s_0, L) = 1/2$. This implies that there exists a policy which is guaranteed to obtain at least reward $1/2$. When taking action $a = R$ at $s_0$, we deterministically transition into a depth-H completely balanced binary tree. We can further constrain the MDP so that the optimal value is $1/2$ (coming from left most branch), though, as we see later, this is not needed.

The feature construction of $\phi \in \mathbb{R}^d$ is as follows: For $s_0, L$, we have $\phi(s_0, L) = e_1$ and $\phi(s_0, R) = e_2$, and $\phi(s_1, a) = e_3$ for any $a \in \{L, R\}$, where $e_1, e_2, e_3$ are the standard basis vectors. For all other states $s \notin \{s_0, s_1\}$, we have that $\phi(s, a)$ is constrained to be orthogonal to $e_1$, $e_2$, and $e_3$, but otherwise arbitrary. In other words, $\phi(s, a)$ has the first three coordinates equal to zero for any $s \notin \{s_0, s_1\}$ but can otherwise be pathological.

The intuition behind this construction is that the features $\phi$ are allowed to be arbitrary complicated for states inside the depth-H binary tree, but are uncoupled with the features on the left path. This implies that both PC-PG and any other algorithm do not have access to a good global function approximator.

Furthermore, as discussed in the following remark, these features do not provide a good approximation of the true dynamics as a linear MDP.

**Remark G.1.** (Linear-MDP approximation failure). As the MDP is deterministic, we would need dimension $d = \Omega(2^H)$ in order to approximate the MDP as a linear MDP (in the sense required in [32]). This is due to that the rank of the transition matrix is $O(2^H)$.

However the on-policy nature of PC-PG ensures that there always exists a best linear predictor that can predict $Q^\pi$ well under the optimal trajectory (the left most path) due to the fact that the features on $s_0$ and $s_1$ are decoupled from the features in the rest of the states inside the binary tree. Thus it means that the transfer error is always zero. This is formally stated in the following lemma.

**Corollary G.1** (Corollary of Theorem 4.3). *PC-PG is guaranteed to find a policy with value greater than $1/2 - \epsilon$ with probability greater than $1 - \delta$, using a number of samples that is $O\left(poly(H, d, 1/\epsilon, \log(1/\delta))\right)$. This is due to the transfer error being zero.*

*Proof of Corollary G.1.* The proof involves showing that the transfer error is 0. Specifically, we will show the following: consider any state-action distribution $\rho$, and any policy $\pi$, and any bonus function $b$ with bounded value for all $b(s, a)$. Then there exists $\theta_\star$ as one of the best on-policy fit, i.e., $\theta_\star \in \arg\min_{\theta:\|\theta\|\leq W} \mathbb{E}_{(s,a)\sim\rho} (\theta \cdot \phi(s, a) - (Q^\pi(s, a) - b(s, a)))^2$, such that:

$$\mathbb{E}_{(s,a)\sim d^\star} (Q^\pi(s, a) - b(s, a) - \theta_\star \cdot \phi(s, a))^2 = 0,$$

i.e., the transfer error is zero.

Let us denote a minimizer of $\mathbb{E}_{(s,a)\sim\rho} (\theta \cdot \phi(s, a) + b(s, a) - Q^\pi(s, a))^2$ as $\widetilde{\theta}$. Note that for any of the first three coordinates of $\widetilde{\theta}$, we have that either the distribution $\rho$ does not put mass on states where this coordinate is non-zero, and we can set the corresponding value in $\widetilde{\theta}$ arbitrarily, or it should be set to a specific value that we show next. Specifically, we set $\widetilde{\theta}_1 = Q^\pi(s_0, L) - b(s_0, L) = 1/2 - b(s_0, L)$, $\widetilde{\theta}_2 = Q^\pi(s_0, R) - b(s_0, R)$, and $\widetilde{\theta}_3 = Q^\pi(s_1, a) - b(s_1, a) = -b(s_1, a)$ for any $a \in \{L, R\}$. This ensures that the fitting error on the $(s_i, a)$ for $i \in \{1, 2\}$ and $a \in \{L, R\}$ is zero, while it does not affect the fit anywhere else as the features in all other states and actions are orthogonal by assumption. Now we take any such minimizer of $\mathbb{E}_{(s,a)\sim\rho} (\theta \cdot \phi(s, a) + b(s, a) - Q^\pi(s, a))^2$ as $\widetilde{\theta}$, where the first three coordinates are fixed to the values given above, and denote it as $\theta_\star$.

For $\theta_\star$, we have $\theta_\star \cdot \phi(s_0, a) = Q^\pi(s_0, a) - b(s_0, a)$ for $a \in \{L, R\}$, and $\theta_\star \cdot \phi(s_1, a) = Q^\pi(s_1, a) - b(s_1, a)$ for $a \in \{L, R\}$, thus, we can verify that $Q^\pi(s_0, a) - b(s_0, a) = \theta_\star \cdot \phi(s_0, a)$ and $Q^\pi(s_1, a) - b(s_1, a) = \theta_\star \cdot \phi(s_1, a)$ for $a \in \{L, R\}$. Since $\pi^\star$ only visits $s_0$ and $s_1$, we can conclude that $\mathbb{E}_{(s,a)\sim d^\star} (Q^\pi(s, a) - b(s, a) - \theta_\star \cdot \phi(s, a))^2 = 0$.

With $\varepsilon_{bias} = 0$, we can conclude the proof by recalling Theorem D.1. $\square$

**Intuition for the success of PC-PG.** Since the corresponding features of the binary subtree have no guarantees in the worst-case, PC-PG may not successfully find the best global policy in general.

However, it does succeed in finding a policy competitive with the best policy that remains in the *favorable* sub-part of the MDP satisfying the modeling assumptions (e.g., the left most trajectory in Figure 4). We do note that the feature orthogonality is important (at least for a provably guarantee), otherwise the errors in fitting value functions on the binary subtree can damage our value estimates on the favorable parts as well; this behavior effect may be less mild in practice.

**Delusional bias and challenges with Bellman backup (and Model-based) approaches.**   While we do not explicitly construct algorithm dependent lower bounds in our construction, we now discuss why obtaining guarantees similar to ours with Bellman backup-based (or even model-based) approaches may be challenging with the current approaches in the literature. We are not assuming any guarantees about the quality of the features in the right subtree (beyond the aforementioned orthogonality). Specifically, for Bellman backup-based approaches, the following two observations (similar to those stressed in Lu et al. [42]), when taken together, suggest difficulties for algorithms which enforce consistency by assuming the Markov property holds:

- (Bellman Consistency) The algorithm does value based backups, with the property that it does an exact backup if this is possible. Note that due to our construction, such algorithms will seek to do an exact backup for $Q(s_0, R)$, where they estimate $Q(s_0, R)$ to be their value estimate on the right subtree. This is due to that the feature $\phi(s_0, R)$ is orthogonal to all other features, so a $0$ error, Bellman backup is possible, without altering estimation in any other part of the tree.

- (One Sided Errors) Suppose the true value of the subtree is less than $1/2 - \Delta$, and suppose that there exists a set of features where the algorithm approximates the value of the subtree to be larger than $1/2$. Current algorithms are not guaranteed to return values with one side error; with an arbitrary featurization, it is not evident why such a property would hold.

More generally, what is interesting about the state aggregation featurization is that it permits us to run *any* tabular RL learning algorithm. Here, it is not evident that *any* other current tabular RL algorithm, including model-based approaches, can achieve guarantees similar to our average-case guarantees, due to their strong reliance on how they use the Markov property. In this sense, our work provides a unique guarantee with respect to model misspecification in the RL setting.

**Failure of concentrability-based approaches**   Some of the prior results on policy optimization algorithms, starting from the Conservative Policy Iteration algorithm Kakade and Langford [34] and further studied in a series of subsequent papers [3, 27, 53] provide the strongest guarantees in settings without exploration, but considering function approximation. As remarked in Section 4.1, most works in this literature make assumptions on the maximal density ratio between the initial state distribution and comparator policy to be bounded. In the MDP of Figure 4, this quantity seems fine since the ratio is at most $H$ for the comparator policy that goes on the left path (by acting randomly in the initial state). However, we can easily change the left path into a fully balanced binary tree as well, with $O(H)$ additional features that let us realize the values on the leftmost path (where the comparator goes) exactly, while keeping all the other features orthogonal to these. It is unclear how to design an initial distribution to have a good concentrability coefficient, but PC-PG still competes with the comparator following the leftmost path since it can realize the value functions on that path exactly and the remaining parts of the MDP do not interfere with this estimation.

# H   Auxiliary Lemmas

**Lemma H.1** (Dimension-free Least Square Guarantees). *Consider the following learning process. Initialize $\theta_1 = \mathbf{0}$. For $i = 1, \ldots, N$, draw $x_i, y_i \sim \nu$, $y_i \in [0, H]$, $\|x_i\| \leq 1$;Set $\theta_{i+1} = \prod_{\Theta := \{\theta : \|\theta\| \leq W\}} (\theta_i - \eta_i(\theta_i \cdot x_i - y_i)x_i)$ with $\eta_i = (W^2)/((W + H)\sqrt{N})$. Set $\hat{\theta} = \frac{1}{N}\sum_{i=1}^{N} \theta_i$, we have that with probability at least $1 - \delta$:*

$$\mathbb{E}_{x \sim \nu}\left[\left(\hat{\theta} \cdot x - \mathbb{E}\left[y|x\right]\right)^2\right] \leq \mathbb{E}_{x \sim \nu}\left[\left(\theta^\star \cdot x - \mathbb{E}\left[y|x\right]\right)^2\right] + \frac{R\sqrt{\ln(1/\delta)}}{\sqrt{N}},$$

*with any $\theta^\star$ such that $\|\theta^\star\| \leq W$ and $R = 3(W^2 + WH)$ which is dimension free and only depends on the norms of the feature and $\theta^\star$ and the bound on $y$.*

*Proof.* Note that we compute $\theta_i$ using Projected Online Gradient Descent [73] on the sequence of loss functions $(\theta \cdot x_i - y_i)^2$. Using the projected online gradient descent regret guarantee, we have that:

$$\sum_{i=1}^{N} (\theta_i \cdot x_i - y_i)^2 \leq \sum_{i=1}^{N} (\theta^\star \cdot x_i - y_i)^2 + \underbrace{W(W+H)}_{:=Q} \sqrt{N}.$$

Denote random variable $z_i = (\theta_i \cdot x_i - y_i)^2 - (\theta^\star \cdot x_i - y_i)^2$. Denote $\mathbb{E}_i$ as the expectation taken over the randomness at step $i$ conditioned on all history $t = 1$ to $i-1$. Note that for $\mathbb{E}_i[z_i]$, we have:

$$\mathbb{E}_i \left[ (\theta_i \cdot x - y)^2 - (\theta^\star \cdot x - y)^2 \right]$$
$$= \mathbb{E}_i \left[ (\theta_i \cdot x - \mathbb{E}[y|x])^2 \right]$$
$$\qquad - \mathbb{E}_i \left[ 2(\theta_i \cdot x - \mathbb{E}[y|x])(\mathbb{E}[y|x] - y) - (\theta^\star \cdot x - \mathbb{E}[y|x])^2 + 2(\theta^\star \cdot x - \mathbb{E}[y|x])(\mathbb{E}[y|x] - y)) \right]$$
$$= \mathbb{E}_i \left[ (\theta_i \cdot x - \mathbb{E}[y|x])^2 - (\theta^\star \cdot x - \mathbb{E}[y|x])^2 \right],$$

where we use $\mathbb{E}[\mathbb{E}[y|x] - y] = 0$. Also for $|z_i|$, we can show that for $|z_i|$ we have:

$$|z_i| = |(\theta_i \cdot x_i - \theta^\star \cdot x_i)(\theta_i \cdot x_i + \theta^\star \cdot x_i - 2y_i)| \leq W(2W + 2H) = 2W(W+H).$$

Note that $z_i$ forms a Martingale difference sequence. Using Azuma-Hoeffding's inequality, we have that with probability at least $1 - \delta$:

$$\left| \sum_{i=1}^{N} z_i - \sum_{i=1}^{N} \mathbb{E}_i \left[ (\theta_i \cdot x - \mathbb{E}[y|x])^2 - (\theta^\star \cdot x - \mathbb{E}[y|x])^2 \right] \right| \leq 2W(W+H)\sqrt{\ln(1/\delta)N},$$

which implies that:

$$\sum_{i=1}^{N} \mathbb{E}_i \left[ (\theta_i \cdot x - \mathbb{E}[y|x])^2 - (\theta^\star \cdot x - \mathbb{E}[y|x])^2 \right] \leq \sum_{i=1}^{N} z_i + 2W(W+H)\sqrt{\ln(1/\delta)N}$$
$$\leq 2W(W+H)\sqrt{\ln(1/\delta)N} + Q\sqrt{N}.$$

Apply Jensen's inequality on the LHS of the above inequality, we have that:

$$\mathbb{E} \left( \hat{\theta} \cdot x - \mathbb{E}[y|x] \right)^2 \leq \mathbb{E} \left( \theta^\star \cdot x - \mathbb{E}[y|x] \right)^2 + (Q + 2W(W+H))\sqrt{\frac{\ln(1/\delta)}{N}}.$$

$\qquad\qquad\qquad\qquad\qquad\qquad\qquad\qquad\qquad\qquad\qquad\qquad\qquad\qquad\qquad\qquad\qquad\qquad\square$

**Lemma H.2.** *Consider the following process. For $n = 1, \ldots, N$, $M_n = M_{n-1} + \Sigma_n$ with $M_0 = \lambda \mathbf{I}$ with $\lambda \geq 1$ and $\Sigma_n$ being PSD matrix with eigenvalues upper bounded by 1. We have that:*

$$2 \log \det(M_N) - 2 \log \det(\lambda \mathbf{I}) \geq \sum_{n=1}^{N} Tr\left( \Sigma_i M_{i-1}^{-1} \right).$$

*Proof.* Note that $M_0$ is PD, and since $\Sigma_n$ is PSD for all $n$, we must have $M_n$ being PD as well.

Using matrix inverse lemma, we have:

$$\det(M_{n+1}) = \det(M_n) \det(\mathbf{I} + M_n^{-1/2}\Sigma_{n+1}M_n^{-1/2}).$$

Add log on both sides of the above equality, we have:

$$\log \det(M_{n+1}) = \log \det(M_n) + \log \det(I + M_n^{-1/2}\Sigma_{n+1}M_n^{-1/2}).$$

Denote the eigenvalues of $M_n^{-1/2}\Sigma_{n+1}M_n^{-1/2}$ as $\sigma_1, \ldots, \sigma_d$, we have:

$$\log \det(M_{n+1}) = \log \det(M_n) + \sum_{i=1}^{d} \log (1 + \sigma_i)$$

Note that since $\lambda \geq 1$, we have $\|M_n^{-1/2}\|_2 \leq 1$ which implies that $\sigma_i \leq 1$, and we have $\log(1+x) \geq x/2$ for $x \in [0,1]$. Hence, we have:

$$\log \det(M_{n+1}) \geq \log \det(M_n) + \sum_{i=1}^{d} \sigma_i/2 = \log \det(M_n) + \frac{1}{2}\mathrm{Tr}\left(M_n^{-1/2}\Sigma_{n+1}M_n^{-1/2}\right)$$

$$= \log \det(M_n) + \frac{1}{2}\mathrm{Tr}\left(\Sigma_{n+1}M_n^{-1}\right),$$

where we use the fact that $\mathrm{Tr}(AB) = \mathrm{Tr}(BA)$ and the trace of PSD matrix is the sum of its eigenvalues. Sum over from $n = 0$ to $N$ and cancel common terms, we conclude the proof. $\square$

**Lemma H.3** (Covariance Matrix Concentration). *Given $\nu \in \Delta(\mathcal{S} \times \mathcal{A})$ and $N$ i.i.d samples $\{s_i, a_i\} \sim \nu$. Denote $\Sigma = \mathbb{E}_{(s,a)\sim\nu}\phi(s,a)\phi(s,a)^\top$. Then, with probability at least $1 - \delta$, we have that:*

$$\left| x^\top \left( \sum_{i=1}^{N} \phi(s_i, a_i)\phi(s_i, a_i)^\top / N - \Sigma \right) x \right| \leq \frac{2\ln(8\widehat{d}/\delta)}{3N} + \sqrt{\frac{2\ln(8\widehat{d}/\delta)}{N}},$$

*with $\widehat{d} = Tr(\Sigma)/\|\Sigma\|$ being the intrinsic dimension of $\Sigma$.*

*Proof.* Denote random matrix $X_i = \phi(s_i, a_i)\phi(s_i, a_i)^\top - \Sigma$. First note that the maximum eigenvalue of $X_i$ is less than 1. Also note that $\mathbb{E}[X_i] = 0$ for all $i$.

Denote $V = \sum_{i=1}^{N} \mathbb{E}[X_i^2]$. For $\mathbb{E}[X_i^2]$, we have

$$\mathbb{E}[X_i^2] \preccurlyeq \mathbb{E}[(\phi_i\phi_i^\top)^2] \preccurlyeq \mathbb{E}[\phi_i\phi_i^T] = \Sigma.$$

where the last inequality uses $\|\phi_i\|_2 \leq 1$. Thus, we have:

$$V = \sum_{i=1}^{N} \mathbb{E}[X_i^2] \preccurlyeq N\Sigma,$$

and $\|V\| \leq N$. Note that the intrinsic dimension of $V$ is exactly equal to the intrinsic dimension of $\Sigma$, which by definition is $\widehat{d}$.

Now apply Matrix Bernstein inequality [65] (Theorem 7.7.1), we have that for any $t \geq \sqrt{N} + 1/3$,

$$\Pr\left(\sigma_{\max}(\sum_{i=1}^{N} X_i) \geq t\right) \leq 4\widehat{d}\exp\left(\frac{-t^2/2}{N + t/3}\right).$$

Since $\sigma_{\max}\left(\sum_{i=1}^{N} X_i\right) = N\sigma_{\max}\left(\sum_{i=1}^{N} X_i/N\right)$, we get that:

$$\Pr\left(\sigma_{\max}\left(\sum_{i=1}^{N} X_i/N\right) \geq \epsilon\right) \leq 4\widehat{d}\exp\left(\frac{-\epsilon^2 N/2}{1 + \epsilon/3}\right),$$

for any $\epsilon \geq \frac{1}{\sqrt{N}} + \frac{1}{3N}$. Set $4\widehat{d}\exp(-\epsilon^2 N/((1 + \epsilon/3))) = \delta$, we get:

$$\epsilon = \frac{2\ln(4\widehat{d}/\delta)}{3N} + \sqrt{\frac{2\ln(4\widehat{d}/\delta)}{N}},$$

which is trivially bigger than $1/\sqrt{N} + 1/(3N)$ as long as $d \geq 1$ and $\delta \leq 1$. This concludes that with probability at least $1 - \delta$, we have:

$$\sigma_{\max}\left(\sum_{i=1}^{N} \phi(s_i, a_i)\phi(s_i, a_i)^\top / N - \Sigma\right) \leq \frac{2\ln(4\widehat{d}/\delta)}{3N} + \sqrt{\frac{2\ln(4\widehat{d}/\delta)}{N}}.$$

We can repeat the same analysis for random matrices $\{X_i := \Sigma - (\phi(s_i, a_i)\phi(s_i, a_i)^\top)\}$ and we can show that with probability at least $1 - \delta$, we have:

$$\sigma_{\max}\left(\Sigma - \sum_{i=1}^{N} \phi(s_i, a_i)\phi(s_i, a_i)^\top / N\right) \leq \frac{2\ln(4\widehat{d}/\delta)}{3N} + \sqrt{\frac{2\ln(4\widehat{d}/\delta)}{N}}.$$

Hence, with probability $1 - \delta$, for any $x$, we have:

$$x^\top\left(\Sigma - \sum_{i=1}^{N} \phi(s_i, a_i)\phi(s_i, a_i)^\top / N\right) x \leq \frac{2\ln(8\widehat{d}/\delta)}{3N} + \sqrt{\frac{2\ln(8\widehat{d}/\delta)}{N}},$$

$$x^\top\left(\sum_{i=1}^{N} \phi(s_i, a_i)\phi(s_i, a_i)^\top / N - \Sigma\right) x \leq \frac{2\ln(8\widehat{d}/\delta)}{3N} + \sqrt{\frac{2\ln(8\widehat{d}/\delta)}{N}}.$$

This concludes the proof. $\qquad\qquad\square$

**Lemma H.4** ( Concentration with the Inverse of Covariance Matrix). *Consider a fixed $N$. Given $N$ distributions $\nu_1, \ldots, \nu_N$ with $\nu_i \in \Delta(\mathcal{S} \times \mathcal{A})$, assume we draw $K$ i.i.d samples from $\nu_i$ and form $\widehat{\Sigma}^i = \sum_{j=1}^{K} \phi(s_j, a_j)\phi(s_j, a_j)^\top / K$ for all $i$. Denote $\Sigma^i = \mathbb{E}_{s,a \sim \nu_i} \phi(s, a)\phi(s, a)^\top$ and $\Sigma = \sum_{i=1}^{N} \Sigma^i + \lambda I$ and $\widehat{\Sigma} = \sum_{i=1}^{N} \widehat{\Sigma}^i + \lambda I$ with $\lambda > 0$. Setting $K = 32N^2 \log\left(8N\widehat{d}/\delta\right)/\lambda^2$, with probability at least $1 - \delta$, we have:*

$$\frac{1}{2}x^T (\Sigma + \lambda I)^{-1} x \leq x^T \left(\widehat{\Sigma} + \lambda I\right)^{-1} x \leq 2x^T (\Sigma + \lambda I)^{-1} x,$$

*for all $x$ with $\|x\|_2 \leq 1$, where we denote the intrinsic dimension $\widehat{d} = \max_{i \in [1,\ldots,N]} \operatorname{tr}(\Sigma^i)/\|\Sigma^i\|$.*

*Proof.* Denote $\eta(K) = \frac{2\ln(8N\widehat{d}/\delta)}{3K} + \sqrt{\frac{2\ln(8N\widehat{d}/\delta)}{K}}$. From Lemma H.3, we know that with probability $1 - \delta$, for all $i$, we have:

$$\Sigma^i + \eta(K)\mathbf{I} + (\lambda/N)\mathbf{I} \succeq \widehat{\Sigma}^i + (\lambda/N)\mathbf{I} \succeq \Sigma^i - \eta(K)\mathbf{I} + (\lambda/N)\mathbf{I},$$

which implies that:

$$\Sigma + N\eta(K)\mathbf{I} + \lambda\mathbf{I} \geq \widehat{\Sigma} + \lambda\mathbf{I} \geq \Sigma - N\eta(K)\mathbf{I} + \lambda\mathbf{I},$$

which further implies that:

$$(\Sigma - N\eta(K)\mathbf{I} + \lambda\mathbf{I})^{-1} \succeq \left(\widehat{\Sigma} + \lambda\mathbf{I}\right)^{-1} \succeq (\Sigma + N\eta(K)\mathbf{I} + \lambda\mathbf{I})^{-1},$$

under the condition that $N\eta(K) \leq \lambda$ which holds under the condition of $K$. Let $U\Lambda U^\top$ be the eigendecomposition of $\Sigma$.

$$x^\top \left(\widehat{\Sigma} + \lambda\mathbf{I}\right)^{-1} x - x^\top (\Sigma + \lambda I)^{-1} x \leq x^\top \left((\Sigma + (-N\eta(K) + \lambda)\mathbf{I})^{-1} - (\Sigma + \lambda\mathbf{I})^{-1}\right) x$$

$$= \sum_i \left((\sigma_i + \lambda - N\eta(K))^{-1} - (\sigma_i + \lambda))^{-1}\right)(x \cdot u_i)^2$$

Since $\sigma_i + \lambda \geq 2N\eta(K)$ as $\sigma_i \geq 0$ and $N\eta(K) \leq \lambda/2$, we have that $2(\sigma_i + \lambda - N\eta(K)) \geq \sigma_i + \lambda$, which implies that $(1/2)(\sigma_i + \lambda - K\eta(N))^{-1} \leq (\sigma_i + \lambda)^{-1}$. Hence, we have:

$$x^\top \left(\widehat{\Sigma} + \lambda\mathbf{I}\right)^{-1} x - x^\top (\Sigma + \lambda I)^{-1} x \leq \sum_{i=1}(u_i \cdot x)^2(\sigma_i + \lambda)^{-1} = x^\top (\Sigma + \lambda\mathbf{I})^{-1}x.$$

The analysis for the other direction is similar. This concludes the proof. $\qquad\qquad\square$

# I  Experimental Details

## I.1  Algorithm Implementation

We implemented two versions of the algorithm: one with a reward bonus which is added to the environment reward (shown in Algorithm 4), and one which performs reward-free exploration, optionally followed by reward-based exploitation using the policy cover as a start distribution (shown in Algorithm 5).

Both of these use NPG as a subroutine, which performs policy optimization using the restart distribution induced by a policy mixture $\Pi_{\mathrm{mix}}$. The implementation of NPG is described in Algorithm 6. We sample states from the restart distribution by randomly sampling a roll-in policy from the cover and a horizon length $h'$, and following the sampled policy for $h'$ steps. Rewards gathered during these roll-in steps are not used for optimization. With probability $\epsilon$, a random action is taken at the beginning of the rollout. We then roll out using the current policy being optimized, and use the rewards gathered for optimization. The policy parameters can be updated using any policy gradient method, we used PPO [55] in our experiments.

For all experiments, we optimized the policy mixture weights $\alpha_1, ..., \alpha_n$ at each episode using 2000 steps of gradient descent, using an Adam optimizer and a learning rate of 0.001. All implementations are done in PyTorch [48], and build on the codebase of [57]. Experiments were run on a GPU cluster which consisted of a mix of 1080Ti, TitanV, K40, P100 and V100 GPUs.

---

**Algorithm 4** PC-PG (reward bonus version)

---

1: **Require**: kernel function $\phi : \mathcal{S} \times \mathcal{A} \to \mathbb{R}^d$
2: Initialize policy $\pi_1$ randomly
3: Initialize policy mixture $\Pi_{\mathrm{mix}} \leftarrow \{\pi_1\}$
4: Initialize episode buffer: $\mathcal{R} \leftarrow \emptyset$
5: **for** episode $n = 1, \dots K$ **do**
6:     **for** trajectory $k = 1, \dots K$ **do**
7:         Gather trajectory $\tau_k = \{s_h^{(k)}, a_h^{(k)}\}_{h=1}^H$ following $\pi_n$
8:         $\mathcal{R} \leftarrow \mathcal{R} \cup \{(s_h^{(k)}, a_h^{(k)})\}_{h=1}^H$
9:     **end for**
10:     Compute empirical covariance matrix: $\hat{\Sigma}_n = \sum_{(s,a)\in\mathcal{R}} \phi(s,a)\phi(s,a)^\top$
11:     Define exploration bonus: $b_n(s,a) = \phi(s,a)^\top \hat{\Sigma}_n^{-1} \phi(s,a)$
12:     Optimize policy mixture weights: $\alpha^{(n)} = \mathrm{argmin}_{\alpha=(\alpha_1,...,\alpha_n),\alpha_i\geq 0, \sum_i \alpha_i=1} \log\det\left[\sum_{i=1}^n \alpha_i \hat{\Sigma}_i\right]$
13:     $\pi_{n+1} \leftarrow \mathrm{NPG}(\pi_n, \Pi_{\mathrm{mix}}, \alpha^{(n)}, N_{\mathrm{update}}, r + b_n)$
14:     $\Pi_{\mathrm{mix}} \leftarrow \Pi_{\mathrm{mix}} \cup \{\pi_{n+1}\}$
15: **end for**

---

## I.2  Environments

### I.2.1  Bidirectional Diabolical Combination Lock

The environment consists of a start state $s_0$ where the agent is placed (deterministically) at the beginning of every episode. The action space consists of 10 discrete actions, $\mathcal{A} = \{1, 2, ..., 10\}$. In $s_0$, actions $1-5$ lead the agent to the initial state of the first lock and actions $6-10$ lead the agent to the initial state of the second lock. Each lock $l$ consists of $3H$ states, indexed by $s_{1,h}^l, s_{2,h}^l, s_{3,h}^l$ for $h \in \{1, ..., H\}$. A high reward of $R_l$ is obtained at the last states $s_{1,H}^l, s_{2,H}^l$. The states $\{s_{3,h}^l\}_{h=1}^H$ are all "dead states" which yield 0 reward. Once the agent is in a dead state $s_{3,h}^l$, it transitions deterministically to $s_{3,h+1}^l$; thus entering a dead state at any time makes it impossible to obtain the final reward $R^l$. At each "good" state $s_{1,h}^l$ or $s_{2,h}^l$, a single action leads the agent (stochastically with equal probability) to one of the next good states $s_{1,h+1}^l, s_{2,h+1}^l$. All other 9 actions lead the agent to the dead state $s_{3,h+1}^l$. The correct action changes at every horizon length $h$ and the stochastic nature of the transitions precludes algorithms which plan deterministically. In addition, the agent receives a negative reward of $-1/H$ for transitioning to a good state, and a reward of 0 for transitioning to a

---

**Algorithm 5** PC-PG (reward-free exploration version)

---

1: **Require**: kernel function $\phi : \mathcal{S} \times \mathcal{A} \to \mathbb{R}^d$
2: Initialize policy $\pi_1$ randomly
3: Initialize policy mixture $\Pi_{\mathrm{mix}} \leftarrow \{\pi_1\}$
4: Initialize episode buffer: $\mathcal{R} \leftarrow \emptyset$
5: **for** episode $n = 1, \ldots K$ **do**
6:     **for** trajectory $k = 1, \ldots K$ **do**
7:         Gather trajectory $\tau_k = \{s_h^{(k)}, a_h^{(k)}\}_{h=1}^H$ following $\pi_n$
8:         $\mathcal{R} \leftarrow \mathcal{R} \cup \{(s_h^{(k)}, a_h^{(k)})\}_{h=1}^H$
9:     **end for**
10:     Compute empirical covariance matrix: $\hat{\Sigma}_n = \sum_{(s,a) \in \mathcal{R}} \phi(s,a)\phi(s,a)^\top$
11:     Define exploration bonus: $b_n(s,a) = \phi(s,a)^\top \hat{\Sigma}_n^{-1} \phi(s,a)$
12:     Optimize policy mixture weights: $\alpha^{(n)} = \mathrm{argmin}_{\alpha = (\alpha_1, \ldots, \alpha_n), \alpha_i \geq 0, \sum_i \alpha_i = 1} \log \det \left[ \sum_{i=1}^n \alpha_i \hat{\Sigma}_i \right]$
13:     $\pi_{n+1} \leftarrow \mathrm{NPG}(\pi_n, \Pi_{\mathrm{mix}}, \alpha^{(n)}, N_{\mathrm{update}}, b_n)$
14:     $\Pi_{\mathrm{mix}} \leftarrow \Pi_{\mathrm{mix}} \cup \{\pi_{n+1}\}$
15: **end for**
16: Initialize policy $\pi_{\mathrm{exploit}}$ randomly
17: $\pi_{\mathrm{exploit}} \leftarrow \mathrm{NPG}(\pi_{\mathrm{exploit}}, \Pi_{\mathrm{mix}}, \alpha^{(K)}, N_{\mathrm{update}}, r)$

---

**Algorithm 6** $\mathrm{NPG}(\pi, \Pi_{\mathrm{mix}}, \alpha, N_{\mathrm{update}}, r)$

---

1: **Input** policy $\pi$, policy mixture $\Pi_{\mathrm{mix}} = \{\pi_1, \ldots, \pi_n\}$, mixture weights $(\alpha_1, \ldots, \alpha_n)$, optional reward bonus $b : \mathcal{S} \times \mathcal{A} \to [0,1]$
2: **for** policy update $j = 1, \ldots N_{\mathrm{update}}$ **do**
3:     Sample roll in policy index $j \sim \mathrm{Multinomial}\{\alpha_1, \ldots, \alpha_n\}$
4:     Sample roll in horizon index $h' \sim \mathrm{Uniform}\{0, \ldots, H-1\}$
5:     Sample start state $s_0 \sim P(s_0)$
6:     **for** $h = 1, \ldots, h'$ **do**
7:         $a_h \sim \pi_j(\cdot|s_h), s_{h+1} \sim P(\cdot|s_h, a_h)$
8:     **end for**
9:     **for** $h = h'+1, \ldots, H$ **do**
10:         $a_h \sim \pi(\cdot|s_h)$ ($\epsilon$-greedy if $h = h'+1$)
11:         $s_{h+1}, r_{h+1} \sim P(\cdot|s_h, a_h)$
12:     **end for**
13:     Perform policy gradient update on return $R = \sum_{h=h'}^H r(s_h, a_h)$
14: **end for**
15: Return $\pi$

---

dead state. Therefore, a locally optimal solution is to learn a policy which transitions to a dead state as quickly as possible, since this avoids the $-1/H$ penalty.

States are encoded using a binary vector. The start state $s_0$ is simply the zero vector. In each lock, the state $s_{i,h}^l$ is encoded as a binary vector which is the concatenation of one-hot encodings of $i, h, l$.

One of the locks (randomly chosen) gives a final reward of 5, while the other lock gives a final reward of 2. Therefore, in addition to the locally optimal policy of quickly transitioning to the dead state (with return 0), another locally optimal solution is to explore the lock with reward 2 and gather the reward there. This leads to a return of $V = 2 - \sum_{h=1}^H \frac{1}{H} = 1$, whereas the optimal return for going to the end of lock with reward 5 is $V^\star = 5 - \sum_{h=1}^H \frac{1}{H} = 4$. In order to ensure that the optimal reward is discovered for every lock, the agent must therefore explore both locks to the end. We used Algorithm 5 for this environment.

### I.2.2 Mountain Car

We used the `MountainCarContinuous-v0` OpenAI Gym environment at `https://gym.openai.com/envs/MountainCarContinuous-v0/`. This environment has a 2-dimensional continuous state space and a 1-dimensional continuous action space. We used Algorithm 4 for this environment.

### I.2.3 Mazes

We used the source code from `https://github.com/junhyukoh/value-prediction-network/blob/master/maze.py` to implement the maze environment, with the following modifications: i) the blue channel (originally representing the goal) is set to zero ii) the same maze is used across all episodes iii) the reward is set to be a constant $0$. We set the maze size to be $20 \times 20$. There are $5$ actions: {up, down, left, right, no-op}. We used Algorithm 5 for this environment, omitting the exploitation step.

### I.3 Additional Figures

(a) RND trace during training

(b) PC-PG final trace

Figure 5: **(a)** shows the state visitation frequencies (brighter color depicts higher visitation frequency) when the RND bonus [16] is applied to a policy gradient method throughout training on the above problem. 'Ep' denotes epoch number showing the progress during a single training run. Although the agent manages to explore to the end of one chain (chain 2 in this case), its policy quickly becomes deterministic and it "forgets" to explore the remaining chain, missing the optimal reward. RND obtains the optimal reward on roughly half of the initial seeds. **(b)** panel shows the traces of policies in the policy cover of PC-PG. Together the policy cover provides a near uniform coverage over both chains.

.

Figure 5 shows the traces of policies trained by RNDs and the traces of the policies from the policy cover of PC-PG.

Figure 6 shows the state visitations of the different policies in the policy cover for Mountaincar.

### I.4 Hyperparameters

All methods were based on the PPO implementation of [57]. For the Diabolical Combination Lock and the MountainCar environments, we used the same policy network architecture: a 2-layer fully connected network with 64 hidden units at each layer and ReLU non-linearities. For the Diabolical Combination Lock environment, the last layer outputs a softmax over 10 actions and for Mountain Car the last layer outputs the parameters of a 1D Gaussian. For the Maze environments, we used a convolutional network with 2 convolutional layers (32 kernels of size $3 \times 3$ for the first, 64 kernels of size $3 \times 3$ for the second, both with stride 2), followed by a single fully-connected layer with

Figure 6: State visitations of different policies in PC-PG's policy cover on MountainCar.

$512$ hidden units, and a final linear layer mapping to a softmax over the $5$ actions. In all cases the RND network has the same architecture as the policy network, except that the last linear layer mapping hidden units to actions is removed. We found that tuning the intrinsic reward coefficient was important for getting good performance for RND. Hyperparameters are shown in Tables 1 and 2.

Table 1: PPO+RND Hyperparameters for Combolock and Mountain Car

| Hyperparameter | Values Considered | Final Value (Combolock) | Final Value (Mountain Car) |
|---|---|---|---|
| Learning Rate | $10^{-3}, 5 \cdot 10^{-4}, 10^{-4}$ | $10^{-3}$ | $10^{-4}$ |
| Hidden Layer Size | 64 | 64 | 64 |
| $\tau_{\text{GAE}}$ | 0.95 | 0.95 | 0.95 |
| Gradient Clipping | 5.0 | 5.0 | 5.0 |
| Entropy Bonus | 0.01 | 0.01 | 0.01 |
| PPO Ratio Clip | 0.2 | 0.2 | 0.2 |
| PPO Minibatch Size | 160 | 160 | 160 |
| PPO Optimization Epochs | 5 | 5 | 5 |
| Intrinsic Reward Normalization | true, false | false | false |
| Intrinsic Reward coefficient | $0.5, 1, 10, 10^2, 10^3, 10^4$ | $10^3$ | $10^3$ |
| Extrinsic Reward coefficient | 1.0 | 1.0 | 1.0 |

Table 2: PPO+RND Hyperparameters for Mazes

| Hyperparameter | Values Considered | Final Value |
|---|---|---|
| Learning Rate | $10^{-3}, 5 \cdot 10^{-4}, 10^{-4}$ | $10^{-3}$ |
| Hidden Layer Size | 512 | 512 |
| $\tau_{\text{GAE}}$ | 0.95 | 0.95 |
| Gradient Clipping | 0.5 | 0.5 |
| Entropy Bonus | 0.01 | 0.01 |
| PPO Ratio Clip | 0.1 | 0.1 |
| PPO Minibatch Size | 128 | 128 |
| PPO Optimization Epochs | 10 | 10 |
| Intrinsic Reward Normalization | true, false | true |
| Intrinsic Reward coefficient | $1, 10, 10^2, 10^3, 10^4$ | $10^3$ |

The hyperparameters used for PC-PG are given in Tables 3 and 4. For the Diabolical Combination Lock experiments, we used a kernel $\phi(s, a) = s$, where $s$ is the binary vector encoding the state described in Section I.2.1. For Mountain Car, we used a Random Kitchen Sinks kernel [50] with 10 features using the following implementation: `https://scikit-learn.org/stable/modules/generated/sklearn.kernel_approximation.RBFSampler.html`. For the Maze environments, we used a randomly initialized convolutional network with the same architecture as the RND network as a kernel.

Table 3: PC-PG Hyperparameters for Combolock and Mountain Car

| Hyperparameter | Values Considered | Final Value (Combolock) | Final Value (MountainCar) |
|---|---|---|---|
| Learning Rate | $10^{-3}, 5 \cdot 10^{-4}, 10^{-4}$ | $10^{-3}$ | $5 \cdot 10^{-4}$ |
| Hidden Layer Size | 64 | 64 | 64 |
| $\tau_{\text{GAE}}$ | 0.95 | 0.95 | 0.95 |
| Gradient Clipping | 5.0 | 5.0 | 5.0 |
| Entropy Bonus | 0.01 | 0.01 | 0.01 |
| PPO Ratio Clip | 0.2 | 0.2 | 0.2 |
| PPO Minibatch Size | 160 | 160 | 160 |
| PPO Optimization Epochs | 5 | 5 | 5 |
| $\epsilon$-greedy sampling | $0, 0.01, 0.05$ | 0.05 | 0.05 |

Table 4: PC-PG Hyperparameters for Mazes

| Hyperparameter | Values Considered | Final Value |
|---|---|---|
| Learning Rate | $10^{-3}, 5 \cdot 10^{-4}, 10^{-4}$ | $5 \cdot 10^{-4}$ |
| Hidden Layer Size | 512 | 512 |
| $\tau_{\text{GAE}}$ | 0.95 | 0.95 |
| Gradient Clipping | 0.5 | 0.5 |
| Entropy Bonus | 0.01 | 0.01 |
| PPO Ratio Clip | 0.1 | 0.1 |
| PPO Minibatch Size | 128 | 128 |
| PPO Optimization Epochs | 10 | 10 |
| $\epsilon$-greedy sampling | 0.05 | 0.05 |