[Reviews · NeurIPS 2020]

Review 1

Summary and Contributions: The paper presents EPOC, a provably convergent policy gradient algorithm for tabular and linear MDPs. Contrary to previous work on policy gradient EPOC explicitly handles the exploration issue and is convergent without strong assumptions on the starting state distribution. The authors provide a PAC bound for EPOC and show that it is robust to a misspecified state aggregation. The last section provide an empirical evaluation of EPOC against PPO and PPO + RND and show that EPOC leads to better results on some hard exploration environments.

Strengths: This paper introduce one of the first provably convergent policy gradient algorithm. This is an important result given that known convergent model-free reinforcement learning algorithms are value based. EPOC also does not require a strong assumption on the starting distribution. The authors provide a sample complexity result for their algorithm. This is a loose bound that may be refined by futurs algorithms, however this is an important result that fills a gap in reinforcement learning theory that has been opened for a long time.

Weaknesses: Policy gradient algorithms are predominantly used over value based methods in environments with continuous state action spaces however the analysis of EPOC requires the action space to be discrete. Could the analysis be extended to continuous action spaces as it has been done for value based algorithm? [1] While the main focus of this work is on the theory side it would be nice to see some experiments with EPOC on larger state action spaces, e.g Mujoco or Atari environments. [1] Efficient Model-free Reinforcement Learning in Metric Spaces, Song & Sun, 2019.

Correctness: I was not able to do a detailed review of the appendix and cannot comment on the correctness of the PAC bound.

Clarity: I enjoyed reading this paper. I am not really familiar with the literature on theory of policy gradient methods but I found the paper easy to follow. The authors made an effort to highlight the limitation of previous work and how their work addresses these issues. small typos L107: \theta and \mu have not been defined? L111: one parenthesis missing in the denominator

Relation to Prior Work: Related work is clearly discussed, including concurrent work from Cai et al. and the relationship between both papers.

Reproducibility: Yes

Additional Feedback:


Review 2

Summary and Contributions: The paper introduces a provably efficient variant of policy gradient algorithm with PAC guarantees for linear function approximation in RKHS setting. The paper shows that under some strong assumption on the MDP including the ability of reset to the initial state, it is possible to learn epsilon-optimal policy in polynomial time by using a variant of natural policy gradient algorithm. The main ideas for making PG PAC is 1- using covariance exploration bonus analogous to the exploration bonus used in linear bandit 2-Changing the initial distribution of policy gradient to the mixture of distributions of prior policies (learned policy cover) to encourage the agent to go to states which where not reachable by previous policies from the getgo. Thus at every iteration the agent tries to expand its span of state visit with respect to the previous iteration. This approach for encouraging the exploration of new states is quite intuitive and is reminiscent of the way KL-regularized RL algos like POLITEX encourage exploration.

Strengths: The paper provides theoretical PAC guarantees for a new variant of policy-gradient method. Normally We tend to assess the quality of a work like this based on the restrictiveness of setting or the strength of assumptions or the tightness of the bounds. In this case one can argue that the theoretical results of this paper is far from a what is expected from a top theoretical work because the assumptions are too restrictive or the access to reset button is too unrealistic and of-course the double digit polynomial PAC bound is probably not useful and far from tight. However, and despite all these technical issues, I believe this is a strong work since it proposes a practical variant of policy-gradient which can deal systematically with the exploration-exploitation trade-off in RL and enjoy PAC performance guarantees in various settings. Especially the fact that the resulting algorithm is very simple and the intuition behind the concept of learned policy cover is quite elegant, make me optimistic (yes optimism in the face of uncertainty!) that this algo can be scaled up using SOTA PG optimizers and produce competitive performance. (As the preliminary experimental result in this paper provides some indication of that.)

Weaknesses: As I mentioned above in terms of technical results there is much room for improvement in terms of the tightness of PAC bounds and the strength of assumptions to achieve this result. But this I don't believe should be a deciding factor for evaluating this work.

Correctness: The theoretical results, to the extent that I could check, seems correct and Also I couldn't see any issue with the experiment methodology and comparison with the prior work have been conducted thoroughly.

Clarity: The paper is written quite clearly and the theoretical analysis and mathematical derivation has been done with rigor. This has been complemented by additional theoretical results which provide insight on behavior of EPOC in the presence of function approximation.

Relation to Prior Work: The state-of-the art has been summarized nicely in Table 1 as much as it is relevant to this paper. A missing reference which I think is quite relevant and close in nature to this work is the POLITEX algo (Yadkordi et al).

Reproducibility: Yes

Additional Feedback:


Review 3

Summary and Contributions: The paper proposes EPOC, a provably effective model-free policy gradient algorithm. It provides the sample complexity of EPOC under a setting of linear MDPs.

Strengths: I think this paper provides a nice theoretical foundation for policy gradient methods.

Weaknesses: The paper contains some typos in proofs.

Correctness: Looks correct.

Clarity: Yes.

Relation to Prior Work: Yes, but I think including a comparison of EPOC to POLITEX would be nice.

Reproducibility: Yes

Additional Feedback: I could check only the proof of Lemma B3, so I might be missing something. According to Theorem C.2, based on which Theorem 3.2 and 3.6 are proven, the sample complexity of EPOC seems to be extremely bad. Why is it so bad? (I guess it is because EPOC's data sampling strategy wastes many samples.) POLITEX algorithm in "POLITEX: Regret Bounds for Policy Iteration Using Expert Prediction" by Yadkori et al is, I think, EPOH without an exploration bonus. What are the differences between EPOC and POLITEX? How can theoretical results in the paper be compared to those in the POLITEX paper? At line 638, it is said that \eta \leq 1/W, but Lemma B2 just assumes \eta=\sqrt{\log(A) / T} / W, which may be larger than 1/W. Am I missing some other assumption? The following comments just point out some typos in proofs. Please fix them. In Eq.6 in Appendix B1, K^n is defined as a set of state-action pairs whose bonus is 0. However, the line below Eq.6 says K^n is a set of state-action pairs whose bonus is positive. According to the definition of K^n in Algorithm 2 and the proof of Lemma B2, the former seems to be correct. In Eq.7, r(s, a^\dagger) and b^n (s, a^\dagger) are undefined. From line 627, I could guess that they are 0, though. At line 643, there must be (1-\gamma) in front of \sum_{t=1}^T (V - V). ----- After Reading the Author Feedback ----- I think the paper provides a nice theoretical foundation for PG methods. As I and Reviewer 2 mentioned, the bound itself is not tight. However, I do not think it is a critical weakness: initial theoretical results tend to be weak, but follow-up papers improve it based on it. My little concern was that differences between EPOC and POLITEX were not clearly explained. However, the authors addressed it very well in the feedback. Therefore, it has been resolved now.

[Author Response · NeurIPS 2020]

We thank the reviewers for carefully reviewing our paper and providing constructive feedback.

**Responses to Reviewer 1.** Our analysis assumes discrete action space but the bound scales as $\log(A)$, which means that in theory can handle large action spaces. Continuous action spaces can also be handled as long as the KL divergence of the initial action distribution (e.g.,$\pi^0(.|s)$) and the comparator's action distribution (e.g., $\pi^\star(.|s)$) is bounded. We will elaborate this point in the revised version.

**Responses to Reviewer 2.** We agree with the reviewer that our current PAC bound is probably not tight and there is much room to potentially improve the bound. Part of the reason is that our approach is on-policy and model-free which makes the dependence on parameters worse than those methods which re-use off-policy data, to perform either LSVI or model-based VI, for instance.

The upshot of our on-policy model-free approach is the robustness to modeling errors that we establish, such as Theorem 3.6 for the classical problem of imperfect state aggregation (see Theorem 3.6). So we believe that there is a trade-off between the best bounds under strong modeling assumptions and more broadly robust techniques. Getting the best of both using a single method is a fascinating direction for future research. We thank the reviewer for recognizing the high-level motivation behind this work though, which is the development of a theoretically sound approach amenable to use in conjunction with practical deep learning and PG methods.

Regarding reset: unlike most policy optimization approaches' analysis, we only assume that we can reset to a fixed initial state (results extend to resetting to a fixed initial distribution), which in our perspective, is equivalent to the common episodic finite horizon setting where agent is also reset at the end of each episode.

**Responses to Reviewer 3.** We are happy to include a more detailed discussion of our approach versus POLITEX (indeed we have already done so in our revision). Note that POLITEX does not explicitly address exploration. Instead it assumes *every* policy is able to visit every state (e.g., Assumption 4 in POLITEX) which is a strong assumption for the underlying MDPs. In other words, POLITEX is not a PAC algorithm for general tabular MDPs and is similar to other works we cite such as [1, 9, 22, 32, 54] in that respect, while EPOC is.

Regarding the sample complexity of EPOC, we agree with the reviewer that there is some data waste. One of the reasons is that we are aiming for a model-free and on-policy algorithm, which potentially wastes samples (as we do not re-use off-policy data from previous rounds), but we get more robustness result under model-misspecification, as we demonstrated in the classic state-abstraction setting (Theorem 3.6). See also response to R2 above.

Thanks for pointing out the slide issue with the learning rate in B.3. We typically assume the number of iterations $T$ is large and at least no smaller than $\log(A)$, which we will clarify in the revised version.

**Additional clarifications for the final version:** A few clarifications and corrections are worth explicitly mentioning to avoid any ambiguities, which we will include in the final version. The critic estimation (Line 6 Alg 2) in the final version will be revised so that $Q^\pi(s, a; r + b^n)$ will be changed to to $Q^\pi(s, a; r + b^n) - b^n$ because this is a convenient change for the special case of linear MDPs, as the shifted $Q$-value is always linear function of $\phi(s, a)$ (since the reward as well as Bellman backup of *any function* are always linear in $\phi(s, a)$), without any need to augment the features as we current did for linear MDPs. The change does not affect the sample complexity or algorithmic properties for the other cases which we study (such as state aggregation, tabular results, and the general agnostic result). This also adresses a minor misspecification in statement of the algorithm (and proof) for linear MDP special case, due to that the current known set definition may be ambiguous to the reader.

[Meta-Review · NeurIPS 2020]

The reviewers unanimously appreciated the paper. The author response's clarified some of their concerns, in particular about POLITEX. Please incorporate the reviewers' feedback into your revisions.